# Demystifying Numerosity in Diffusion Models—Limitations and Remedies

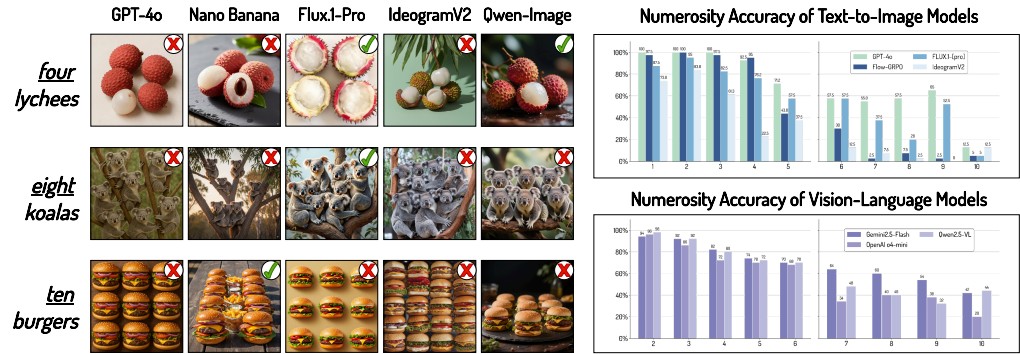

Figure 1: **Key Motivations**: On the left, we display typical images generated with the latest text-to-image models, most of which fail to adhere to numerosity instructions. On the right, we illustrate how numerosity poses a challenge for both text-to-image generation models and vision-language models.

## Abstract

Numerosity remains a challenge for state-of-the-art text-to-image generation models like FLUX and GPT-4o, which often fail to accurately follow counting instructions in text prompts. In this paper, we aim to study a fundamental yet often overlooked question: *Can diffusion models inherently generate the correct number of objects specified by a textual prompt simply by scaling up the dataset and model size?* To enable rigorous and reproducible evaluation, we construct a clean synthetic numerosity benchmark comprising two complementary datasets: GrayCount250 for controlled scaling studies, and NaturalCount6 featuring complex naturalistic scenes. Second, we empirically show that the scaling hypothesis does not hold: larger models and datasets alone fail to improve counting accuracy on our benchmark. Our analysis identifies a key reason: diffusion models tend to rely heavily on the noise initialization rather than the explicit numerosity specified in the prompt. We observe that noise priors exhibit biases toward specific object counts. In addition, we propose an effective strategy for controlling numerosity by injecting count-aware layout information into the noise prior. Our method achieves significant gains, improving accuracy on GrayCount250 from 20.0% to 85.3% and on NaturalCount6 from 74.8% to 86.3%, demonstrating effective generalization across settings.

## 1 Introduction

Despite impressive advancements in large language models and photorealistic image generation, the seemingly elementary skill of numerosity still trips them up. The limitation first went viral as the *strawberry test*, where pre-reasoning LLMs repeatedly failed the trivial query "How many 'r's are in the word 'strawberry'?", a task many models inexplicably answered with 2 instead of the correct 3. Several recent efforts Fu et al. (2024); Xu & Ma (2024); Yehudai et al. (2024) have studied these limitations from different aspects, including tokenization, architecture, and the training dataset. This numerosity blind spot extends to image generation: state-of-the-art diffusion models such as FLUX Labs (2024), IdeogramV2 Ideogram (2024), GPT-4o image OpenAI (2025), Nano Banana Google AI (2024) and Qwen-Image Wu et al. (2025) struggle to produce the correct number

of objects specified in text prompts (see Figure 1), where results are evaluated on the GeckoNum benchmark Kajić et al. (2024). Whether counting characters or visual objects, basic quantitative reasoning remains a persistent Achilles' heel across generative models.

In this paper, we focus on the numerosity of representative visual generative models, a.k.a. diffusion models, as their inability to generate accurate object counts limits applications requiring numeric fidelity, from scientific illustration to data-driven storytelling. Our study starts with a fundamental question: can diffusion models inherently generate the correct number of objects specified by a textual prompt simply by scaling up the dataset and model size? We position this study as a stress test of the "scaling hypothesis" Esser et al. (2024); Snell et al. (2024); Hoffmann et al. (2022); Bi et al. (2024); Yin et al. (2024); Kaplan et al. (2020) , examining whether merely increasing model size, training data and compute budget is sufficient to eliminate counting errors in generated images.

One fundamental challenge in studying numerosity scaling hypothesis is the inherent diversity and noisiness of real-world images. Occlusions, clutter, and ambiguous boundaries lead even state-of-the-art vision–language models (e.g., OpenAI o4-mini OpenAI (2025) and Gemini2.5-Flash Ideogram (2025)) to miscount, making reliable ground-truth annotations difficult to obtain, as shown in Figure 1. Humans face a similar hurdle: our fast, intuitive System 1 provides only a rough "gist" and can subitize small sets, but precise enumeration requires deliberate System 2 processing to index and track items. Without this effort, counts above four become unreliable. Therefore, text-to-image diffusion models must learn to bind numeric tokens to discrete spatial slots for accurate object generation.

Motivated by the inherent challenge of collecting real-world images with accurate counts, we propose curating two synthetic numerosity datasets: (1) GrayCount250, a high-quality scalable synthetic numerosity dataset using paste-layer-to-layout scheme with transparent objects on gray backgrounds. This design enables the synthesis of scalable and count-accurate image–text pairs, facilitating controlled scaling studies; and (2) NaturalCount6, a naturalistic synthetic numerosity dataset focusing on 2-6 objects in complex real-world scenes with rigorous count verification. This dual approach allows us to systematically study numerosity under both controlled and naturalistic conditions. We also perform numerosity probing to evaluate the counting discrimination capability of our trained models with a diffusion classifier scheme.

Our controlled experiments reveal that the scaling hypothesis fails for diffusion models: even when scaling from 2B to 12B model and expanding the dataset from 1K to 500K samples, performance consistently degrades as the target count increases. A deeper analysis uncovers a key underlying cause: diffusion models primarily optimize with respect to the noise prior, rather than the text instructions, making object count a weak signal easily overridden by stochastic noise patterns. Surprisingly, we find that different noise priors exhibit distinct preferred numerosity modes, constraining counting ranges even when prompted with varying object counts. In other words, current text-to-image diffusion models lack effective mechanisms to bind textual numerals to spatial instance slots, leading to systematic counting errors.

To leverage this noise dependency, we propose a set of simple yet effective, count-aware noise conditioning techniques that steer the noise prior toward layouts consistent with the requested number of instances. Empirically, these strategies achieve substantial improvements: on GrayCount250, uniform scaled noise prior dramatically improves overall accuracy from 20.0% to 85.3%, with particularly strong performance on higher counts (>30: from 3.81% to 75.0%); on NaturalCount6, Gaussian noise prior boosts overall accuracy from 74.83% to 86.33%, with notable gains for challenging higher counts (count-5: from 56.67% to 72.55%). These results not only underscore the decisive role that 2-D noise priors play in accurate counting, but also demonstrate successful generalization across both controlled and naturalistic settings. Taken together, we provide both diagnostic insights and practical solutions for improving numerosity in text-to-image generation. We hope that the community will reconsider the fundamental limitations of diffusion models in following spatially aware instructional prompts. Our key contributions are summarized as follows:

- A scalable, high-quality, numerosity-oriented data engine that curates hundreds of thousands of image–text pairs for reproducible evaluation.
- Rigorous experiments demonstrating that the scaling hypothesis fails for diffusion models on numerosity, with analysis revealing a key cause: strong dependency on the noise prior.
- A count-aware noise conditioning techniques that achieve substantial improvement on GrayCount250 and NaturalCount6 dataset by leveraging the diffusion model's inherent bias toward its noise prior.

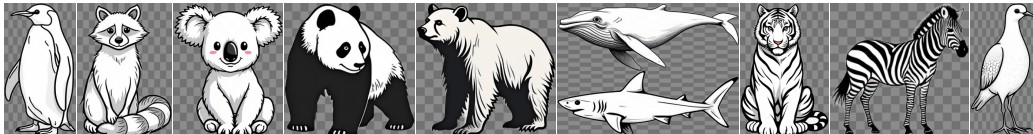

Figure 2: **Pipeline for Constructing the GrayCount250 Dataset.** We first prepare a list of concepts, and combine these concept names with a transparent object prompt for the FLUX.1-[dev]. Then, we apply RMBG 2.0 to perform matting and obtain the transparent object layers. Synthetic numerosity images are generated by pasting transparent object layers onto random layouts corresponding to target counts.

Figure 3: Illustration of the transparent object layers used to construct the GrayCount250 dataset. All layers are rendered in a minimalist black line art style for simplicity.

## 2 RELATED WORK

Numerosity is a key metric for evaluating the capabilities of visual perception and generation models. Evaluations on well-known numerosity-related benchmarks, such as T2ICountBench Cao et al. (2025), GenEval Ghosh et al. (2023), and GeckoNum Kajić et al. (2024), consistently show performance degradation beyond five objects. Even state-of-the-art models still struggle to exceed 50% accuracy on single-digit prompts. However, these studies primarily focus on *revealing the existence* of numerosity failures in small number ranges (1-10), without systematically investigating whether scaling data, model size, or compute can resolve these limitations and the challenge of evaluation and annotation.

The existing efforts broadly fall into three paths: (1) iterative editing with external analyzers Binyamin et al. (2024); Wu et al. (2024), (2) instance-level spatial guidance via attention masks Feng et al. (2022) or location tokens Wang et al. (2024); Epstein et al. (2023), and (3) direct model optimization through reward-driven fine-tuning Gill (2024); Zafar et al. (2024); Liu et al. (2025); Xue et al. (2025). Recent work has explored the connection between noise initialization and image layout Hsueh et al. (2025); Mao et al. (2023); Ban et al. (2024); Mao et al. (2024). Li et al. (2025) and Samuel et al. (2024) demonstrate that searching for optimal noise seeds can improve compositional generation or rare concepts generation, while INITNO Guo et al. (2024) optimizes noise via cross-attention guidance. However, these approaches require thousands of generations to identify suitable noise patterns, making them computationally prohibitive for practical applications.

## 3 SYNTHETIC NUMEROSITY BENCHMARK CONSTRUCTION

Text-to-image numerosity generation requires datasets that support reliable annotation and evaluation. Real-world images, however, often suffer from imprecise annotations due to occlusion and clutter. To address this, first, we explain how to construct a high-quality, counting-oriented synthetic numerosity dataset with precise annotations and minimal labor cost, designed for controlled scaling studies. Then, we construct a naturalistic numerosity dataset with complex scenes, focusing on a narrower count range to ensure annotation reliability. Building on this foundation, we introduce two evaluation methods: (i) quantitative numerosity metrics to systematically measure counting errors in generated images, and (ii) a zero-shot diffusion classifier to assess whether diffusion models can accurately count the number of objects in a given image.

### 3.1 HIGH-QUALITY SCALABLE SYNTHETIC NUMEROSITY DATASET : GRAYCOUNT250

We illustrate the entire dataset construction pipeline in Figure 2. We sample keywords from a list of more than 250 concepts and apply a predefined template prompt (see Template B.1) to FLUX.1-[dev]Labs (2024) to generates images with solid-colored backgrounds. RMBG-2.0AI (2024) then extracts foregrounds and alpha masks. To ensure high visual quality, we apply human filtering

Figure 4: Illustration of the synthetic numerosity dataset. **GrayCount250** (left): We show randomly arranged images containing 8 koalas, 15 pandas, 38 crocodiles, and 43 sea lions. **NaturalCount6** (right): We show images with 3 avocados, 4 soccer balls, 5 wallets, and 6 remote controls with prompts provided in Appendix B.

and retrain the top 250 transparent layers (see Figure 3). This number was chosen to balance object diversity and scalability, covering a wide range of semantic concepts (e.g., animals, foods, flowers).

To study numerosities from 1 to 50, we employ a random layout strategy on a $512 \times 512$ canvas, repeatedly placing the high-quality transparent layers according to sampled object centers to achieve target counts. Figure 4 shows examples of this layout with different object counts. The training set includes subsets of 1K, 5K, 50K, 100K, and 500K samples, constructed from 20, 100, 100, 100, and 250 concepts respectively, with each concept uniformly spanning all counts (1 to 50). This approach requires diffusion models to reason about spatial relationships during generation, closely mimicking real-world counting scenarios.

For evaluation, we use a held-out category (rabbit) absent from all training subsets. It spans the 1–50 numerosity range with 50 prompts. For each target count, we generate 10 images and estimate the actual object numbers using CountGD Amini-Naieni et al. (2024). We additionally report accuracies on multiple other held-out categories and observe comparable results, indicating that the evaluation set is not tied to a specific concept. We select *rabbit* because its detection is the most reliable, with CountGD achieving 99.78% exact accuracy on this set (see Appendix D).

### 3.2 Naturalistic Synthetic Numerosity Dataset : NaturalCount6

We construct a naturalistic dataset focusing on counts from 2 to 6 objects, where real-world annotations are particularly challenging due to occlusion and clutter. We select 24 diverse object categories, including animals, fruits, toys, and household items, and use GPT-4o to generate varied, context-rich prompts for each target (see Template B.3). To ensure high visual quality and count accuracy, we employ a cascaded generation pipeline (SD-XL primary, SD-3.5 backup, FLUX fallback). Each generated image is rigorously filtered using CountGD to retain only those with exact count matches, resulting in a balanced training set of 17,509 images across 16 categories, with 8 held-out for testing (see examples in Figure 4). For evaluation, all images undergo both CountGD detection and manual verification to ensure reliability in complex scenes with potential occlusion or ambiguity.

### 3.3 Evaluation Metrics

**Measuring Numerosity Accuracy.** Given a dataset of $N$ images, let $r_i$ denote the number of objects requested in the $i$-th prompt, and $p_i$ the predicted count on the generated image. We evaluate the counting performance with three complementary metrics: $\text{ExactAccuracy(EAcc. )} = \frac{1}{N} \sum_{i=1}^{N} \mathbb{I}(p_i = r_i)$, which measures strict, perfect enumeration; $\text{MeanAbsoluteError(MAE)} = \frac{1}{N} \sum_{i=1}^{N} |p_i - r_i|$, which quantifies the average magnitude of counting errors; and $\text{ToleranceAccuracy(TAcc. )} = \frac{1}{N} \sum_{i=1}^{N} \mathbb{I}(|p_i - r_i| \leq T)$, assessing approximate correctness within a tolerance threshold $T$ ($T = 2$ unless otherwise specified).

**Probing Numerosity Perception.** We study whether diffusion model can perceive the number of the objects presented in a given image. Inspired by the Diffusion Classifier Li et al. (2023), we transform a pre-trained diffusion model into a zero-shot numerosity classifier as shown in Figure 5. Given an input image $x$, we determine which prompt best explains the image by selecting the count $k$ that minimizes the diffusion loss $L_k(x)$ between predicted and actual noise. Assuming a uniform prior over counts, this approach is equivalent to: $\hat{k} = \arg\max_k p(k \mid x) \propto \arg\min_k L_k(x)$, where $L_k(x)$ is computed using the prompt "A photo of {k target objects}." To balance efficiency and accuracy, we use the coarse-to-fine approach described in Figure 5. The specific parameters of this strategies (e.g., the size of the candidate pool and timestep allocation) were chosen based on an empirical analysis of the trade-offs involved (see Appendix E).

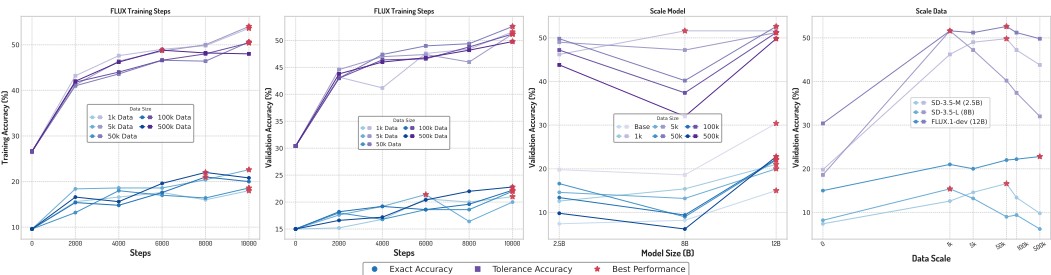

Figure 5: Our proposed coarse-to-fine numerosity classifier. Given an input image $x$ and a set of possible counts $k \in \{1, 2, ..., 50\}$, we choose the conditioning $c$ ("A photo of $k$ rabbits.") that best predicts the noise added to the input image. To balance efficiency and accuracy, we employ a two-phase strategy: a *Coarse Screening* phase evaluates all 50 counts with 50 timesteps to identify the top 20 candidates, followed by a *Refinement* phase that uses 200 timesteps on these candidates to determine the final count.

Figure 6: Performance trends across scaling dimensions. From left to right: (a) training accuracy over steps for FLUX models trained on datasets of different sizes (1K–500K samples); (b) corresponding validation accuracy; (c) effect of model scale on counting accuracy; (d) effect of dataset scale. **Key findings** include low overall accuracy, performance degradation with larger models or excessive data, and limited gains from extended training.

Table 1: Mean Absolute Error across different count ranges for SD-3.5-Medium, SD-3.5-Large, and FLUX.1-dev trained with various data size. Lower values indicate better performance, with best results highlighted in green .

| Data Size | SD-3.5-M | | | | | SD-3.5-L | | | | | FLUX.1-dev | | | | |
|---|---|---|---|---|---|---|---|---|---|---|---|---|---|---|---|
| | 1-10 | 10-20 | 20-30 | >30 | Overall | 1-10 | 10-20 | 20-30 | >30 | Overall | 1-10 | 10-20 | 20-30 | >30 | Overall |
| – | 1.20 | 5.57 | 14.3 | 31.3 | 17.3 | 1.08 | 4.81 | 11.1 | 25.4 | 14.0 | 0.511 | 4.00 | 8.82 | 26.5 | 13.8 |
| 1k | 0.978 | 2.18 | 3.49 | 8.50 | 4.88 | **0.811** | **1.75** | 4.61 | 6.25 | 4.04 | 0.333 | 1.71 | **3.11** | 7.80 | 4.30 |
| 5k | **0.756** | **1.90** | 3.49 | **7.53** | 4.38 | 0.956 | 3.13 | **3.97** | **5.09** | 3.73 | **0.233** | 1.98 | 3.41 | 7.57 | 4.30 |
| 50k | 0.922 | 2.18 | **3.43** | 8.05 | 4.67 | 1.23 | 3.63 | 4.71 | 5.19 | 4.07 | 0.244 | 1.73 | 3.30 | 7.66 | 4.27 |
| 100k | 1.17 | 2.10 | 3.46 | 8.00 | 4.68 | 0.811 | 3.79 | 5.56 | 7.51 | 5.21 | 0.289 | 1.71 | 3.45 | **7.05** | 4.05 |
| 500k | 1.08 | 2.17 | 4.13 | 7.96 | 4.80 | 1.37 | 4.32 | 5.73 | 7.09 | 5.23 | 0.267 | **1.38** | 3.25 | 8.48 | 4.53 |

## 4 NUMEROSITY SCALING HYPOTHESIS FOR DIFFUSION MODELS

Although prior work has established that diffusion models struggle with numerosity tasks Ghosh et al. (2023); Conwell et al. (2024); Kajić et al. (2024), it remains an open question whether they can inherently learn to generate the correct number of objects through scaling. This section systematically investigates if the *scaling hypothesis*—that increasing model and dataset size improves performance—holds for numerosity generation.

A major obstacle to this inquiry is the lack of suitable real-world datasets. Natural images suffer from (1) data scarcity and long-tailed count distributions, being heavily biased toward low counts (e.g., 1-5 objects), and (2) unreliable evaluation due to occlusion, clutter, and annotation noise, making accurate supervision labor-intensive.

To quantify the reliability gap, we conducted a controlled comparison (see Appendix D.3) evaluating two counting methods on natural versus synthetic images. The results show a substantial performance drop on natural image, underscoring the challenge of using in-the-wild data for clean scaling studies. This motivates our use of the GrayCount250 datasets introduced in Section 3.1.

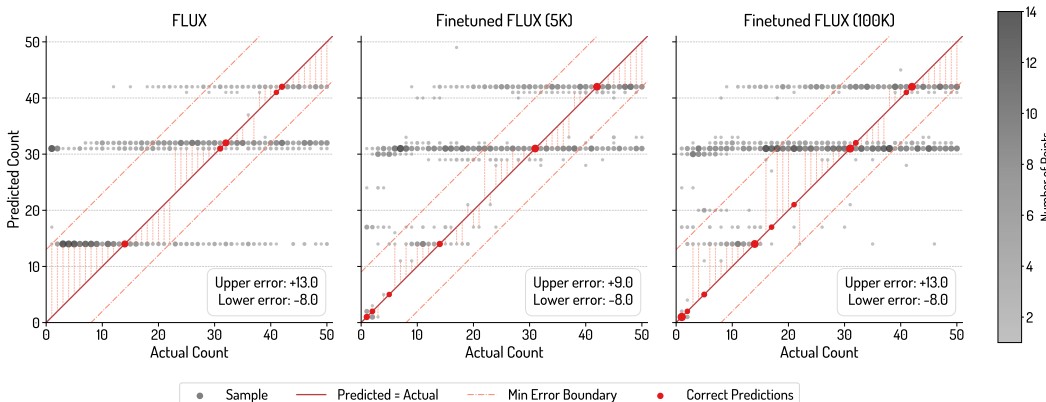

Figure 7: Predicted vs. actual count scatter plots comparing vanilla FLUX.1-dev (left) with models fine-tuned on 5K (center) and 100K (right) synthetic controlled datasets. **Key findings:** Base model exhibits strong prediction clustering, treating numerosity as coarse categories (low counts ∼14, high counts ∼30) ; fine-tuning improves precision mainly for small counts (<10) but fails to generalize to higher quantities.

## 4.1 Scaling is Not Enough: Limited Gains and Diminishing Returns

For our investigation, we finetune 3 state-of-the-art models: Stable-Diffusion-3.5-Medium (2.5B) stabilityai (2024), Stable-Diffusion-3.5-Large (8B) stabilityai (2024), and FLUX.1-dev (12B) Labs (2024). Each model is trained for 10K steps with early stopping, using LoRA (rank=16) with batch size (64) and resolution (512), trained on GrayCount250 ranging from 1K to 500K samples.

**Our systematic evaluation reveals non-trivial failures in numerosity learning for diffusion models, even under idealized conditions.** Contrary to expectations, models exhibit poor performance on both training and validation sets (panels a and b of Figure 6), with exact accuracy remaining below 23% across all dataset configurations. This indicates a fundamental limitation in diffusion models' numerosity perception, beyond data scale.

**Scaling model size, data, or training steps fails to yield consistent improvements.** As shown in panel c of Figure 6, increasing model capacity fail to systematically enhance counting accuracy. FLUX shows slight gains, likely due to its pretraining regime rather than scale. Panel d reveals that data scaling beyond 5K samples leads to performance degradation. Furthermore, extended training only improves tolerance accuracy, bringing negligible gains in exact counting (panels a and b). These results collectively suggest that the numerosity bottleneck is inherent to the diffusion architecture.

## 4.2 Diffusion Models are Weak Numerosity Classifiers

To understand the nature of these limitations, we examine how diffusion models perceive and classify numeric quantities. Following the same approach as our training set, we generate a set of 1,000 images (counts 1-50) using a held-out category ("rabbits"). Then, we apply numerosity probing to diffusion models trained on different synthetic controlled dataset and summarize the classification performance when tasked with numerosity perception in Figure 7.

**Base Models Perceive Numerosity Coarsely.** Base model (left) groups predictions around specific values, treating numerosity as broad categories. This indicates an inherent tendency to distinguish "few" from "many" without precise counting.

**Fine-tuning Fails to Generalize Beyond Small Counts.** Fine-tuned models (center, right) show improved precision for counts below 10, with tighter error bounds, yet fail to generalize to larger quantities. This reveals that diffusion models lack the structural capacity for exact numerosity reasoning across a wide range.

Overall, our analysis demonstrates that scaling is insufficient for numerosity tasks. The consistently low accuracy on both training and validation sets under simple synthetic conditions indicates a fundamental limitation in diffusion architectures. Since models fail to achieve precise counting even in controlled settings, their performance in complex real-world scenarios would face greater challenges.

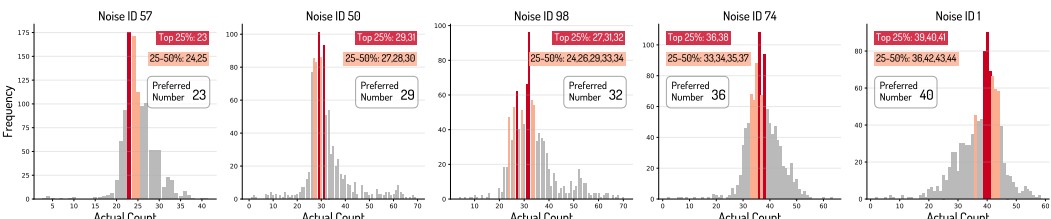

Figure 8: Count distributions across different noise initializations with the same requested counts in the range of 30-50 objects. Each noise prior creates a distinct preferred number, with 50% of all images clustering around specific values (red/orange bars). This demonstrates that initial noise strongly determines the number of objects generated regardless of text instructions.

# 5 Demystifying Numerosity in Diffusion Models

In this section, we turn to a more fundamental question: what specific component in diffusion models most strongly influences and constrains numerosity generation? Modern text-to-image diffusion models adopt the Rectified Flow framework, which can be formalized as:

$$\mathbf{x}_t = (1-t)\mathbf{x}_0 + t\boldsymbol{\epsilon}, \quad \frac{d\mathbf{x}_t}{dt} = \mathbf{v}_\theta(\mathbf{x}_t, t, c), \quad \boldsymbol{\epsilon} \sim \mathcal{N}(0, \mathbf{I}) \tag{1}$$

where $\mathbf{x}_0$ is the generated image, $\boldsymbol{\epsilon}$ is the noise sampled from a standard Gaussian distribution, $\mathbf{x}_t$ represents the interpolated state at timestep $t$, $\mathbf{v}_\theta$ is the velocity field predicted by the model, and $c$ is the text conditioning. Both the initial noise $\boldsymbol{\epsilon}$ and the text conditioning $c$ potentially affect the number of objects generated in the final image.

In Section 5.1 , we analyze how these two components interact, revealing that noise initialization dominates the counting process. Then Section 5.2 and Section 5.3 demonstrate how this insight can be leveraged to better understand and control numerosity in diffusion models.

## 5.1 The Role of Noise in Numerosity Generation

Inspired by prior work Li et al. (2025) showing that initial noise correlates with object arrangement, we systematically investigate whether and how this relationship extends to numerosity. We conducted a large-scale study to assess how noise and text interact during and after training. Using a FLUX.1-dev model fine-tuned on 5K GrayCount250, we sample 100 noise vectors and pair each with 3,000 diverse counting prompts (1–50 counts, 30 categories), generating 300,000 images to systematically evaluate training's effect on noise-text interplay. We provide more results and analyses in Appendix F–G.

**Different Noise Priors Determine Distinct Counting Ranges.** As shown in Figure 8, different noise priors consistently bias the generated numerosity toward specific values, often overriding the text instruction. This preference varies widely across noise seeds, indicating that initial noise exerts a stronger influence on the final count than the prompt.

**Noise Determines Layout, Text Activates Locations.** Figure 9 shows that for a fixed noise prior, objects are consistently generated in similar positions across different prompts. This indicates that the noise strongly determines the spatial layout, while the text prompt primarily influences which objects appear at those pre-determined locations, rather than accurately controlling their quantity. Training amplifies this dependence, as encoding layout in noise proves easier than learning text-based counting. This inherent bias explains why scaling data or model fails to overcome counting limitations.

## 5.2 Validating the Dominant Role of Noise through Noise Control

To test whether noise dominates numerosity generation, we design controlled experiments including three complementary noise manipulation methods that explicitly inject layout cues into the initial noise. All methods operate in latent space and modify noise within target bounding boxes $\text{box}_j (j \in [1, N])$ to assess how structured noise influences counting accuracy:

- **Uniform Scaled**: Scales noise inside boxes by a factor $\gamma$, creating sharp boundaries.

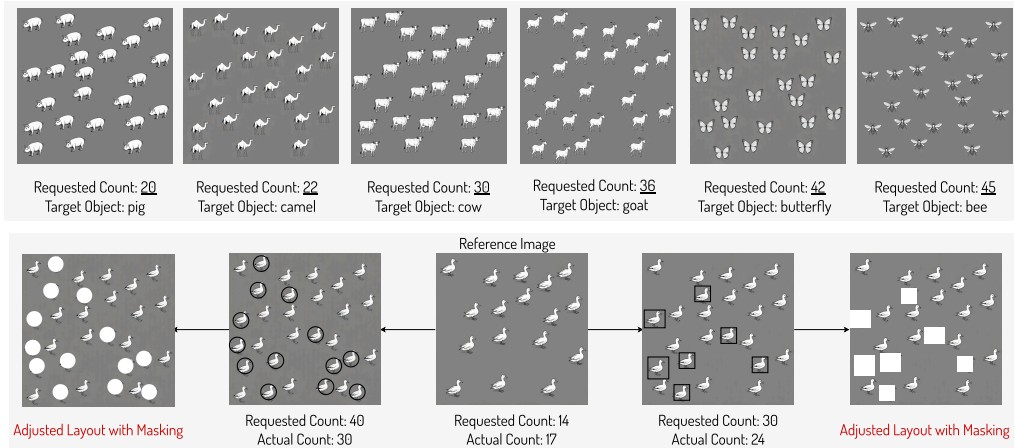

Figure 9: Illustration of how the noise prior determines the overall layout arrangement. Top: given a fixed noise prior, we observe that the diffusion model consistently generates objects in nearly identical positions and even tends to produce the exact same number of target objects, despite under different prompts. Bottom: we show that images with different object counts—generated from the same noise—share similar layouts. The center image can be transformed into left/right variants by masking, as objects occupy nearly identical positions.

- **Fixed**: Replaces box noise with a fixed sample $\mathbf{z}^*$ to enforce consistency.
- **Gaussian**: Adds a Gaussian kernel centered at each box: $\mathbf{z}_t(\mathbf{x}) = \boldsymbol{\epsilon}(\mathbf{x}) + \sum_j w \exp\left[-\frac{1}{2}(\mathbf{x} - \boldsymbol{\mu}_j)^\top \boldsymbol{\Sigma}_j^{-1}(\mathbf{x} - \boldsymbol{\mu}_j)\right]$, where $\boldsymbol{\mu}_j$ is the box center and $\boldsymbol{\Sigma}_j = \mathrm{diag}(\sigma_j^2, \sigma_j^2)$ with $\sigma_j = \alpha|\mathrm{box}_j|_2$ controlling the influence area; $w$ and $\alpha$ are fixed during training for simplicity.

Table 2: Performance comparison across different priors for different count ranges: 1-10, 10-20, 20-30, and >30. Best performance within each range for each metric is highlighted in **bold**.

| Prior Type | **Exact Accuracy (%) ↑** | | | | | **Tolerance Accuracy (%) ↑** | | | | | **Mean Absolute Error ↓** | | | | |
|---|---|---|---|---|---|---|---|---|---|---|---|---|---|---|---|
| | 1-10 | 10-20 | 20-30 | >30 | Overall | 1-10 | 10-20 | 20-30 | >30 | Overall | 1-10 | 10-20 | 20-30 | >30 | Overall |
| *Baseline* | 77.8 | 19.0 | 3.00 | 3.81 | 20.0 | **100** | 75.0 | 46.0 | 21.4 | 51.2 | 0.23 | 1.98 | 3.41 | 7.57 | 4.30 |
| *Noise Prior* | | | | | | | | | | | | | | | |
| Gaussian | 86.7 | 62.0 | 30.0 | 12.4 | 39.2 | 98.9 | 99.0 | 84.0 | 59.1 | 79.2 | 0.16 | 0.49 | 1.23 | 2.79 | 1.54 |
| Fixed | **95.6** | **97.0** | 87.6 | 63.8 | 81.0 | 97.8 | **100** | 99.0 | 93.2 | 96.6 | 0.16 | **0.03** | 0.15 | 0.60 | 0.32 |
| Uniform Scaled | 87.8 | 94.9 | **96.2** | **75.0** | **85.3** | **100** | **100** | **100** | **98.9** | **99.5** | **0.12** | 0.05 | **0.04** | **0.34** | **0.19** |
| *Prompt Prior* | | | | | | | | | | | | | | | |
| Instance-wise | 83.3 | 30.0 | 8.0 | 7.6 | 25.8 | **100** | 72.0 | 53.0 | 23.3 | 53.8 | 0.17 | 1.83 | 2.98 | 6.95 | 3.91 |

## 5.3 EXPERIMENTAL RESULTS

We conduct comprehensive experiments to validate our noise dominance hypothesis across two settings: FLUX.1-dev model finetuned on our 5K GrayCount250 dataset, and FLUX.1-dev model finetuned on NaturalCount6 dataset for generalization validation. We use the same training settings as in Section 4. Here, *baseline* refers to the model trained without any noise prior.

**Performance of Different Priors on GrayCount250 Dataset.** Table 2 demonstrates that all noise-based methods dramatically outperform baseline training on GrayCount250 dataset, confirming our hypothesis about noise dominance in numerosity generation. We set $\omega = 0.3, \alpha = 0.8$ for Gaussian method, while using intensity multiplier $\gamma = 0.1$ for Uniform Scaled approach (detailed ablation studies on parameters and image quality analyses are presented in Appendix I). The Fixed approach excels at lower count ranges, while Uniform Scaled performs better at higher counts due to the decreasing relative area occupied by each object as counts increase. To validate that noise rather than text controls counting, we tested instance-wise prompts that explicitly enumerate each object (e.g., "k target objects: object 1: description; ... object k: description"). This approach showed only modest improvements over baseline, mainly in lower count ranges. This reinforces our observation that text prompts have limited influence over spatial arrangement even when explicitly detailing each object.

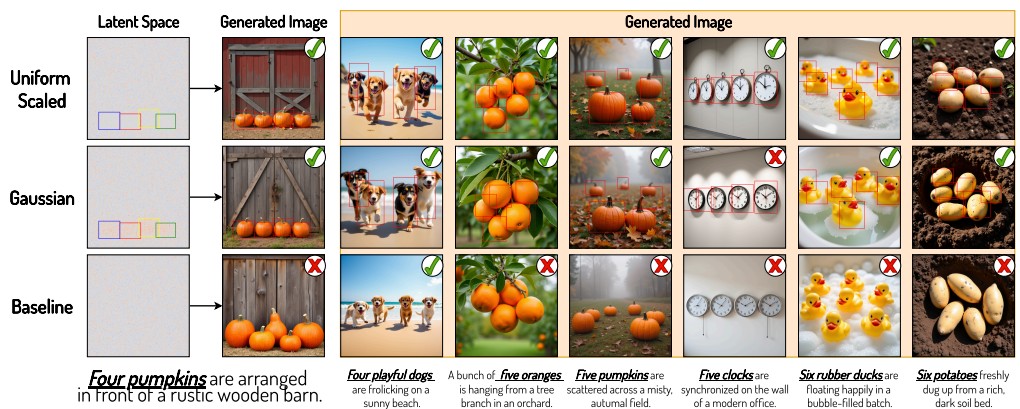

Figure 10: Numerosity generation on NaturalCount6 validation set. Left: latent space structure of noise priors. Right: results for noise prior methods and baseline (training w/o prior).

Table 3: Performance comparison on naturalistic images and training dynamics. Noise prior methods show superior performance, especially for higher counts, and continuous improvement during training.

| Method | Exact Accuracy by Count (%) | | | | | | Overall Exact Accuracy at Steps (%) | | | | |
| --- | --- | --- | --- | --- | --- | --- | --- | --- | --- | --- | --- |
| | 2 | 3 | 4 | 5 | 6 | Overall | 2K | 4K | 6K | 8K | 10K |
| FLUX.1-dev | 93.97 | 88.24 | 83.33 | 52.50 | 44.44 | 72.47 | – | – | – | – | – |
| Baseline | 95.83 | 80.83 | 80.00 | 56.67 | 60.83 | 74.83 | 74.67 | **74.83** | 74.00 | 74.37 | 74.00 |
| *Noise Prior* | | | | | | | | | | | |
| Uniform Scaled | 91.67 | 87.16 | 88.99 | **78.26** | **72.73** | 85.00 | 76.67 | 80.17 | **85.00** | 83.50 | 84.33 |
| Gaussian | **97.20** | **92.73** | **92.23** | 72.55 | 65.00 | **86.33** | 73.67 | 76.50 | 81.00 | 83.67 | **86.33** |

**Performance of Noise Priors on NaturalCount6 Dataset.** As shown in Table 3 and Figure 10, both Gaussian and Uniform Scaled methods achieve substantial improvements over baseline training across all count ranges, with particularly notable gains for challenging higher counts. The training dynamics reveal a critical insight: while baseline training quickly plateaus and struggles to progress further, noise prior methods demonstrate relatively continuous improvement throughout training. These results demonstrate that our core findings about noise prior dominance generalize beyond line-art style settings, providing a practical foundation for improving numerosity control in naturalistic text-to-image generation applications.

We conducted additional analyses on generalization capabilities, including extended count ranges (1–100), object size variation, and category diversity effects, as detailed in Appendix J. Additionally, we explored alternative strategies to mitigate noise bias, but all proved significantly less effective than direct noise modification (see Appendix K). This result underscores a critical limitation: diffusion models encode spatial information primarily in the noise prior rather than learning robust text-to-count mappings, a weakness acutely exposed in precise numerosity tasks.

## 6 CONCLUSION

This paper challenges the conventional scaling hypothesis on the fundamental task of generating the correct number of objects in an image with diffusion models. We rigorously show that merely increasing diffusion model capacity or training data volume fails to improve and can even degrade numerosity performance, on a scalable, high-quality, synthetic, and numerosity-oriented dataset. These surprising results motivate a deeper analysis, revealing that diffusion models determine their layout decisions on noise initialization rather than on the count specified in the text prompt. Leveraging this insight, we explicitly inject numerosity information into the noise prior, achieving substantially better results. We hope our findings inspire the community to rethink the inherent limitations of diffusion models in counting and other reasoning tasks.

**Reproducibility and Ethics Statements** This work presents fundamental research on diffusion models' numerosity capabilities using synthetic datasets free from sensitive content. We release anonymous code at `https://anonymous.4open.science/r/Numer-FLUX-7F48` and supplementary material to support reproducibility. Experimental details are provided in Sections 4–5, with datasets and metrics defined in Section 3. Additional implementations and extended results are included in the Appendix to support reproducibility.

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

# APPENDIX

## A NUMEROSITY ACCURACY OF STATE-OF-THE-ART MODELS

**Text-to-Image Models.** To evaluate the performance of image generation models (GPT-4o OpenAI (2025) , Gemini2.0-flash Ideogram (2025), FLUX.1-[pro] Labs (2024), IdeogramV2 Ideogram (2024), SimpleAR Wang et al. (2025)) and Flow-GRPO Liu et al. (2025) on counting tasks, we selected the numeric-simple subtask from the GeckoNum Kajić et al. (2024) benchmark, as its counting range is larger compared to GenEval Ghosh et al. (2023) (which only covers 1-4). Additionally, due to the complexity of real counting images, conventional detection models struggle to obtain accurate answers. Therefore, GeckoNum results require manual annotation, and all results in the following table are manually annotated. More specifically, the numeric-simple subtask includes 80 object classes for counting 1-5 and 40 object classes for counting 6-10, resulting in a total of 600 samples.

Table 4: GeckoNum Benchmark Performance: exact accuracy (%) of GPT-4o, FLUX.1-[pro], IdeogramV2, SD3.5-M+Flow-GRPO and SimpleAR-1.5B-RL on the numeric-simple subtask (Object Counts: 1–10).

| Model | # Object Count | | | | | | | | | | Avg |
|---|---|---|---|---|---|---|---|---|---|---|---|
| | 1 | 2 | 3 | 4 | 5 | 6 | 7 | 8 | 9 | 10 | |
| GPT-4o | 100.00 | 100.00 | 100.00 | 92.50 | 71.25 | 57.50 | 55.00 | 57.50 | 65.00 | 12.50 | **71.13** |
| FLUX.1-[pro] | 87.50 | 95.00 | 82.50 | 76.25 | 57.50 | 57.50 | 37.50 | 20.00 | 52.50 | 5.00 | 57.13 |
| IdeogramV2 | 73.75 | 83.75 | 61.25 | 22.50 | 37.50 | 12.50 | 7.50 | 2.50 | 0.00 | 12.50 | 31.38 |
| SD3.5-M+Flow-GRPO | 97.50 | 100.00 | 97.50 | 95.00 | 43.75 | 30.00 | 2.50 | 7.50 | 2.50 | 5.00 | 48.13 |
| SimpleAR-1.5B-RL | 42.50 | 21.25 | 16.25 | 5.00 | 3.75 | 0.00 | 0.00 | 2.50 | 0.00 | 0.00 | 9.12 |

**Vision-Language Models.** The numerosity task is challenging not only for text-to-image generation models but also for Vision Language Models (VLMs) when detecting the number of objects in images, as it requires understanding the target object categories and accurately reasoning about spatial relationships. Therefore, obtaining accurate results is particularly challenging. To evaluate counting accuracy of VLMs on real-world images, we created a controlled dataset with object counts ranging from 2 to 10 across 50 common object categories. The dataset construction methodology is detailed in the supplementary materials. Each image is annotated with a ground-truth count, and we evaluated three state-of-the-art VLMs: Gemini2.0-Flash, OpenAI o4-Mini, and Qwen2.5-VL Bai et al. (2025). For consistent evaluation, we used a standardized prompt template: "How many {object} are there in the image? Output the counts in JSON format: {'count': <number>}". The comparative counting performance across these VLMs is visualized in Figure 11, which demonstrates their respective counting results.

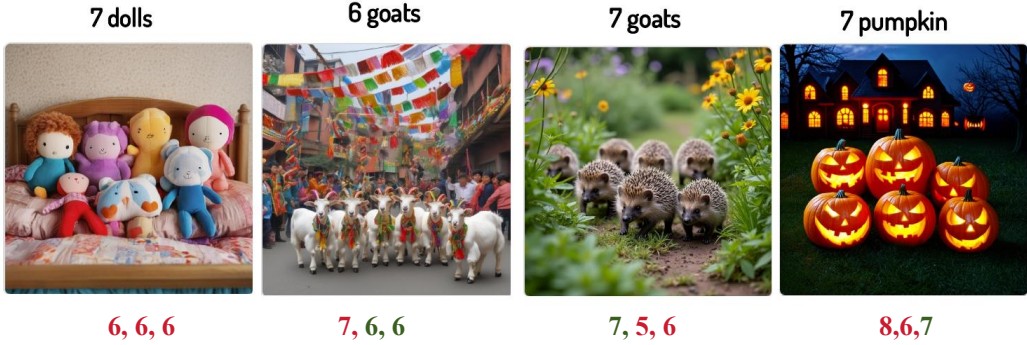

Figure 11: Examples on numerosity accuracy of Vision-Language Models. Numbers shown below each image represent predictions from (left to right): Gemini2.0-Flash, OpenAI o4-Mini, and Qwen2.5-VL.

## B Prompt Templates for Dataset Generation and Evaluation

Our synthetic dataset construction and evaluation rely on two carefully designed prompt templates. These templates ensure consistency across all generated images and provide clear instructions to the text-to-image models.

### B.1 Transparent Object Prompt Template

The following template is used in the initial stage of our data pipeline (as described in Section 3) to generate high-quality, isolated objects with FLUX.1-[dev]. This template produces clean, minimalist line art objects with transparent backgrounds, which are then extracted using RMBG-2.0 for our synthetic dataset construction.

---

**Transparent Object Prompt for FLUX.1-[dev]**

Generate an image of a {animal} in a minimalist black line art style. The {animal} should be depicted using clean, uniform black lines with no shading or fill, focusing on its shape and essential features. The lines should be crisp and precise, creating a modern and elegant design. Ensure the overall composition is balanced and visually appealing, isolated on a solid gray background.

---

### B.2 Minimalist Line-Art Prompt Template

After constructing our synthetic numerosity images by placing transparent object layers according to specific layouts, we use the following template variants for both training diffusion models and evaluating their numerosity capabilities. These templates are systematically designed to standardize the visual style while clearly communicating the desired number of objects. Each prompt follows the same structure but offers slight variations to improve model robustness.

---

**Minimalist Line-Art Prompt Template**

- A minimalist black line drawing of {count} {animal}, set against a soft gray background, each outlined with smooth, precise lines to create an elegant and harmonious composition.
- A sleek and simple black line art depiction of {count} {animal}, all drawn with clean, straight lines against a soft gray background, highlighting their graceful yet minimalist form.
- An artistic, minimalist rendering of {count} {animal} in black line art, featuring sharp, clean outlines and a neutral gray backdrop, providing a balanced and sophisticated aesthetic.
- A charming collection of {count} {animal}, each drawn with minimalist black lines, creating a graceful and balanced visual impression on a soft gray canvas.
- A simple yet elegant black line drawing of {count} {animal}, captured in clear, refined lines on a gray background, emphasizing their sleek shapes and natural beauty.
- An understated illustration of {count} {animal} in black line art, gracefully outlined against a soft gray backdrop. The simplicity of the design brings out the elegant beauty of each animal.
- A refined, minimalist black line art of {count} {animal}, with each figure outlined with clean precision against a subtle gray background, creating a serene and balanced visual composition.
- A modern take on black line art, featuring {count} {animal} drawn with smooth and clear lines, set against a soft gray backdrop, exuding simplicity and elegance.
- A clean, minimalist design of {count} {animal} in black line art, placed against a soft gray background, with each animal portrayed in a calm and graceful manner.
- An elegant black line illustration of {count} {animal}, outlined in precise, simple lines, against a smooth gray background, capturing their minimalist charm and beauty.

---

As mentioned in Section 3, we employ these prompt templates consistently across our entire dataset, spanning numerosities from 1 to 50. This standardization allows us to isolate and study the specific effects of numerosity without confounding variables from prompt variations.

## B.3    Naturalistic Dataset Prompt Generation Template

As described in Section 3, we employ GPT-4o to generate diverse and context-rich prompts for each target count. The specific prompt used for GPT-4o is provided below.

> **Minimalist Line-Art Prompt Template**
>
> "Please generate [number] counting prompts for text-to-image models with the following classes: [${class_names}]. Each prompt should contain a number between 2 and 6 and follow this format:
> index.prompt|number of objects|object name
> For example:
> 1.Three dogs are playing with each other|3|dog
> 2.An image of four cars driving down the street.|4|car
> Ensure diversity by varying sentence structures, actions, and environments. Use different verbs, adjectives, and locations to make each prompt unique. The prompts should be reasonable. Return exactly ten prompts, starting from index 1, and separate them by '\n'."

The following prompts were used to generate the naturalistic numerosity dataset images displayed in Figure 4. Each prompt specifies the target object count and places the objects in a realistic, contextually appropriate scene.

- A family of three avocados sits neatly on a wooden kitchen countertop, waiting to be sliced.
- Four soccer balls are scattered across a soccer field during a lively kids' practice session.
- Five wallets are neatly arranged on a display table in a high-end boutique.
- Six remote controls are neatly arranged on a wooden coffee table in a cozy living room.

## C    Random Non-Overlapping Object Placement Algorithm

As described in Section 3, we designed two layout strategies for generating our synthetic numerosity dataset: grid-based and random. Here, we detail the algorithm used to generate our random non-overlapping layouts.

Algorithm 1 outlines our approach, which dynamically resizes objects based on the target count to maintain visual clarity while maximizing placement success rates. The algorithm works as follows:

1. First, we calculate appropriate object sizes based on the total canvas area and target object count.
2. For each object, we attempt to find a non-overlapping position through random sampling.

## D    Extended Evaluation Reliability Results

In this section, we justify our evaluation design. First, we conduct extended evaluations across diverse object categories to assess generalization. Second, we report CountGD's accuracy on our evaluation set to establish measurement reliability. Finally, we analyze counting performance on naturalistic images with cluttered backgrounds to demonstrate the necessity of GrayCount250 for controlled scaling studies.

### D.1    Generalization Across Concept Categories

To ensure a reliable evaluation of numerosity perception while demonstrating method generalization, we conducted extensive analysis across multiple concept categories. Although we selected "rabbit" as the primary held-out category for our benchmark due to its superior detection reliability (as quantified in the following paragraph), we emphasize that the learned numerosity ability generalizes well beyond this specific category.

To validate this claim, we evaluated performance across five diverse held-out categories using the FLUX model trained on 5K samples. As shown in Table 5, performance remains consistent across categories with similar baseline accuracy (approximately 21% without prior). More importantly, the Gaussian prior brings significant and consistent improvements across all categories, confirming that:

---

**Algorithm 1** Random Non-Overlapping Layout Generation

---

**Require:** Animal image $image$, number of animals $n$, canvas size $(w, h)$
**Ensure:** Canvas image with $n$ non-overlapping animals
 1: Create canvas $canvas$ with size $(w, h)$
 2: Dynamically resize animal based on $n$ and canvas area to get $resized\_image$
 3: Get resized animal size $(animal\_w, animal\_h)$
 4: $positions \leftarrow \emptyset$ ▷ Empty list to track placed animals
 5: **for** $i \leftarrow 1$ to $n$ **do**
 6:     $attempt \leftarrow 0$
 7:     $placed \leftarrow$ false
 8:     **while** $attempt < max\_attempts$ and $\neg placed$ **do**
 9:         $x \leftarrow$ random integer in range $[0, w - animal\_w]$
10:         $y \leftarrow$ random integer in range $[0, h - animal\_h]$
11:         **if** position $(x, y, animal\_w, animal\_h)$ doesn't overlap with any in $positions$ **then**
12:             Add $(x, y, animal\_w, animal\_h)$ to $positions$
13:             Paste $resized\_image$ onto $canvas$ at position $(x, y)$
14:             $placed \leftarrow$ true
15:         **end if**
16:         $attempt \leftarrow attempt + 1$
17:     **end while**
18:     **if** $\neg placed$ **then**
19:         **return** null ▷ Unable to place all animals
20:     **end if**
21: **end for**
22: **return** $canvas$

---

Table 5: Generalization performance across five held-out categories. Accuracy (Exact Match, %) is reported with and without Gaussian prior.

| Concept | Accuracy w/o Prior | Accuracy w/ Prior |
|---------|--------------------|--------------------|
| Rabbit | 20.0 | 39.2 |
| Bird | 21.6 | 39.4 |
| Cake | 21.2 | 32.4 |
| Dog | 21.2 | 38.0 |
| Umbrella | 21.2 | 32.0 |

(1) The improvements are not unique to "rabbit" but generalize across semantic and visual concepts;
(2) The proposed method remains effective under varied object types and visual characteristics.

### D.2 DETECTION RELIABILITY OF COUNTGD

The selection of "rabbit" as our primary evaluation category was motivated by its exceptional detection performance using CountGD Amini-Naieni et al. (2024). To quantify this reliability, we created a dedicated test set consistent with our evaluation category (rabbit). We generate a set of 5,000 images, 100 samples for each counts (1-50)—using a single held-out category ("rabbits"), following the same approach as our training set. Table 7 presents the detailed performance metrics of CountGD across different count ranges, the average Exact Accuracy is 99.78% and tolerance Accuracy is 100.00%, showing consistent high accuracy regardless of the number of objects in the images.

### D.3 ANALYSIS OF EVALUATION RELIABILITY ON NATURAL VS. SYNTHETIC IMAGES

To objectively compare the reliability of counting evaluation on different data types, we constructed two dedicated datasets for 10 object categories (*teddy bear, turtle, duck, seal, mug, rabbit, dog, orange, horse, cat*) across counts 2 to 10 (90 images per dataset).

Figure 12: Example images from the two datasets constructed for evaluating counting reliability: naturalistic scenes with cluttered backgrounds (top), and minimalist line-art objects on structured random layouts (bottom).

**Datasets.** Using the pipeline in Section 3, we constructed two datasets: one with naturalistic scenes and cluttered backgrounds, and the other with minimalist line-art objects on structured layouts (see examples in Figure 12).

**Evaluation.** We assessed the counting performance using two methods: CountGD, a model-based counting tool, and Qwen-2.5-VL-7B-Instruct, a general-purpose vision-language model. Exact match accuracy was measured for each dataset.

**Results.** As shown in Table 6, both counting methods exhibited significantly higher accuracy on the synthetic dataset. This performance gap demonstrates that the inherent challenges in naturalistic images introduce substantial noise into the evaluation process, justifying our use of a clean synthetic benchmark for controlled scaling studies.

Table 6: Exact match counting accuracy (%) across the constructed naturalistic and synthetic image sets.

| Image Type | Qwen-2.5-VL | CountGD |
|---|---|---|
| Synthetic (Structured) | 85 | 98 |
| Naturalistic (Cluttered) | 70 | 84 |

## E    DESIGN RATIONALE FOR THE COARSE-TO-FINE NUMEROSITY CLASSIFIER

Our work adapts the Diffusion Classifier framework Li et al. (2023) for the novel task of numerosity classification. The core motivation for the coarse-to-fine strategy is computational feasibility. Exhaustively evaluating all 50 numerosity prompts with 1000 timesteps using FLUX.1-Dev is prohibitively slow. This strategy first inexpensively identifies likely candidates with 50 timesteps before committing to a refined evaluation.

While the original Diffusion Classifier prunes to a top-5 pool, we found this recall (14.3%) too low for fine-grained numerosity estimation. Expanding to top-20 increased recall to 54.1%, better balancing coverage and cost. The use of fewer timesteps (50) for the coarse stage follows the finding in Li et al. (2023) that fewer uniform timestep sampling can still provide a sufficiently reliable signal for ranking candidates.

To further quantify the contribution of each stage, we report detailed accuracies in Table 8. The results highlight the inherent difficulty of the task—Top-1 accuracy remains low. However, the refinement stage is crucial: it significantly improves the average rank of the ground-truth count from approximately 20 to 10, substantially reducing prediction instability despite the modest gain in exact accuracy. This justifies the additional computation, as allocating more resources to likely candidates is key for reliable ranking.

## F    EXTENDED ANALYSIS OF NOISE-DOMINANCE NUMEROSITY GENERATION

This appendix provides extended analyses complementing Section 4. First, we present comprehensive count distributions across a wider range of noise initializations, demonstrating the pervasiveness of

Table 7: Evaluation results of CountGD detection model accuracy across different count ranges.

| Metrics | 1-10 | 10-20 | 20-30 | >30 | Overall |
|---|---|---|---|---|---|
| Sample Size | 900 | 1000 | 1000 | 2100 | 5000 |
| Exact Accuracy (%) | 100.0 | 99.9 | 99.9 | 99.57 | 99.78 |
| Mean Absolute Error | 0.000 | 0.001 | 0.001 | 0.0043 | 0.002 |
| Tolerance Accuracy (±2, %) | 100.0 | 100.0 | 100.0 | 100.0 | 100.0 |

Table 8: Classifier performance before and after refinement

| Model | Stage (Timesteps) | Top1 (%) | Top20 (%) | Avg. Rank | Median Rank |
|---|---|---|---|---|---|
| FLUX | Coarse (50) | 2.2 | 45.8 | 23.05 | 22.00 |
| FLUX | Refine (200) | 2.5 | — | 10.36 | 10.00 |
| Finetuned FLUX (5K) | Coarse (50) | 2.8 | 54.1 | 19.86 | 19.00 |
| Finetuned FLUX (5K) | Refine (200) | 3.3 | — | 10.41 | 10.00 |

noise-driven counting preferences. Second, we validate the consistency of this phenomenon across models trained with different dataset sizes.

### F.1 EXTENDED COUNT DISTRIBUTIONS ACROSS NOISE PRIORS

Building on Figure 8 from Section 5, we now show distributions across 100 noise initializations. The vast majority exhibit strong preferences for specific count ranges, independent of textual instructions. While not all noise vectors show equally strong preferences due to the high-dimensional noise space, the phenomenon remains remarkably consistent. We organize the distributions into three ranges (20-30 in Figure 13, 30-40 in Figure 14, and 40-50 in Figure 15) to illustrate the pervasiveness of this behavior.

### F.2 CROSS-MODEL CONSISTENCY OF NOISE PREFERENCES

To further validate our hypothesis about the dominant role of noise prior in determining object counts, we conduct experiments using two different models: one trained on GrayCount250 5K data and another trained on GrayCount250 100K data. Both models share identical architectures and training settings, differing only in the size of their training datasets.

**Experiment Setup.** Following the setup in Section 4.1, we use the same 100 noise vectors from $\mathcal{N}(0, \mathbf{I})$ as in the main text, combining them with text prompts varying across counts (30-50), object categories (30 types), and templates. For visualization and direct comparison, we focus on the same five representative noise priors shown in Figure 8 of the main text.

**Empirical Observations.** Figure 18 shows the count distributions for five representative noise priors across both models. We observe several key patterns:

1) Each noise prior exhibits a consistent preferred number range across both models, despite their different training data scales (5K vs. 100K).

2) The preferred numbers are highly specific - for instance, Noise ID 57 consistently prefers 23, while Noise ID 1 consistently prefers 40, regardless of the model used.

This consistency reinforces our analysis in Section 4.1 that the initial noise prior fundamentally determines the spatial layout and, consequently, the object count. These preferences persist even with more training data suggests that this is not a training artifact but rather an intrinsic property of how diffusion models process spatial information.

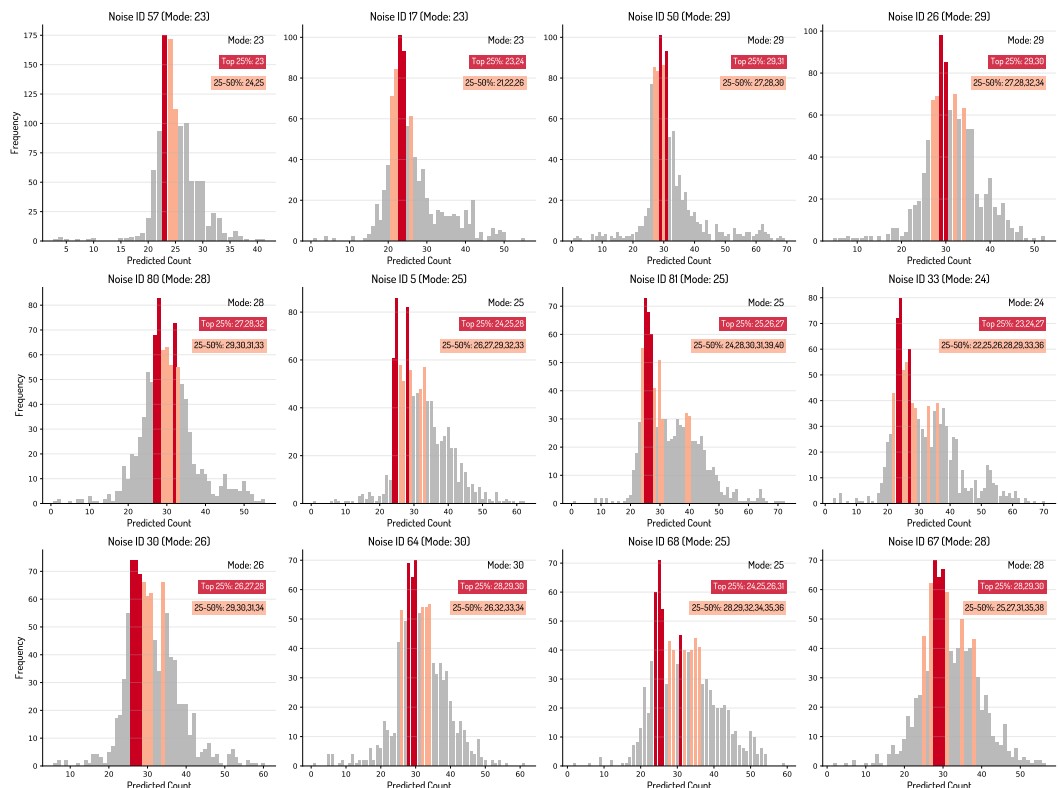

Figure 13: Noise priors with preferred counts in the 20-30 range. Each subplot represents a different noise initialization, showing how it consistently produces counts in this range regardless of the requested count in the prompt.

## G    STATISTICAL ANALYSIS OF OBJECT SPATIAL PATTERNS

In Section 4 of the main text (see Figure 9), we qualitatively demonstrate that, under a fixed noise prior, diffusion models tend to generate objects in highly similar spatial arrangements, regardless of the requested count or object category. This observation supports the hypothesis that "*noise determines layout, text activates locations.*"

Here, we provide a comprehensive quantitative investigation of this phenomenon. Specifically, we analyze the spatial distribution of all generated objects under a fixed noise prior (identical to the one used in Figure 9 of the main text), offering statistical evidence to complement and extend the qualitative findings in the main text. We focus on two aspects: (A.1) the clustering structure of object positions for different actual counts, and (A.2) the evolution and stability of cluster centers as the actual count increases.

### G.1    CLUSTERING OF OBJECT POSITIONS FOR DIFFERENT ACTUAL COUNTS

As demonstrated in Section 4, we fix the noise prior and systematically vary the text prompt—including the requested count (1–50), object category (30 animal types), and template (2 variants)—to generate a total of 50 × 30 × 2 = 3,000 images. Due to the stochastic nature of the generative process, the actual object count in each image does not perfectly match the requested count, resulting in a non-uniform distribution of actual counts across the dataset.

Given all generated images under this same noise prior, we first apply CountGD Amini-Naieni et al. (2024) (with the specified object category) to detect bounding boxes in each image. For every image, this yields a set of bounding boxes, their center coordinates, and the corresponding actual count (i.e., the number of detected objects). We then group all images by their actual count $n$, aggregate the center coordinates $\{\mathbf{c}_i\}_{i=1}^N$ within each group, and perform K-means clustering with $k = n$ for each

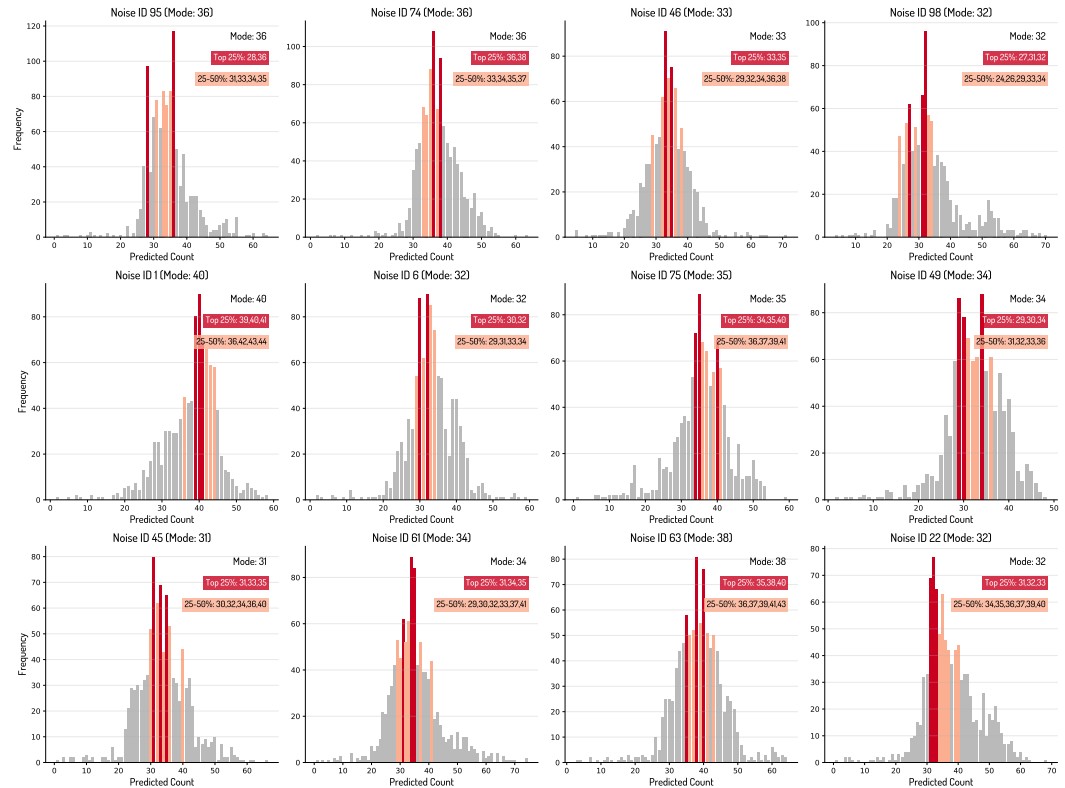

Figure 14: Noise priors with preferred counts in the 30-40 range.

group:

$$\min_{\{\mu_j\}} \sum_{i=1}^{N} \min_{j \in [1,n]} \|\mathbf{c}_i - \mu_j\|^2 \tag{2}$$

where $\mu_j$ denotes the $j$-th cluster center.

Because the clustering structure in images with small actual counts is not obvious, and the number of images with large counts is limited, we focus our visualization on the range $n = 20$ to $n = 30$. In this regime, the sample size is sufficient and the spatial patterns are most representative. As shown in Figure 16, object positions cluster into clear and consistent patterns. The cluster centers are highly regular, directly confirming that the noise prior determines the spatial layout.

## G.2 EVOLUTION AND STABILITY OF CLUSTER CENTERS

To quantify how the spatial template changes as $n$ increases, we match cluster centers between consecutive counts using optimal assignment (Hungarian algorithm). For cluster centers $\{\mu_j^{(n)}\}_{j=1}^{n}$ and $\{\mu_j^{(n+1)}\}_{j=1}^{n+1}$, we solve:

$$\min_{\pi} \sum_{j=1}^{n} \|\mu_j^{(n)} - \mu_{\pi(j)}^{(n+1)}\| \tag{3}$$

where $\pi$ is a permutation. We define the stability score as the fraction of matched centers with distance less than a threshold $\tau$:

$$\text{Stability}(n, \tau) = \frac{1}{n} \sum_{j=1}^{n} \mathbb{I}(\|\mu_j^{(n)} - \mu_{\pi(j)}^{(n+1)}\| < \tau) \tag{4}$$

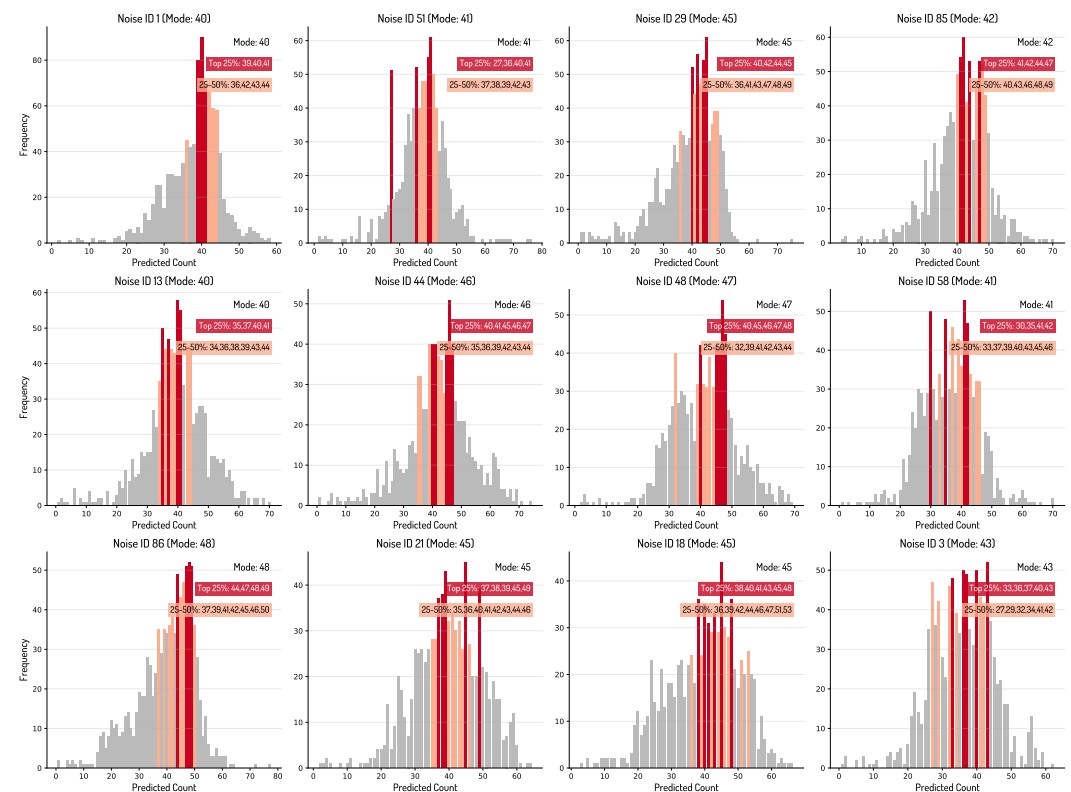

Figure 15: Noise priors with preferred counts in the 40-50 range. Even at these higher numbers, some initialization maintains its characteristic distribution pattern, further confirming that noise dominates numerosity control.

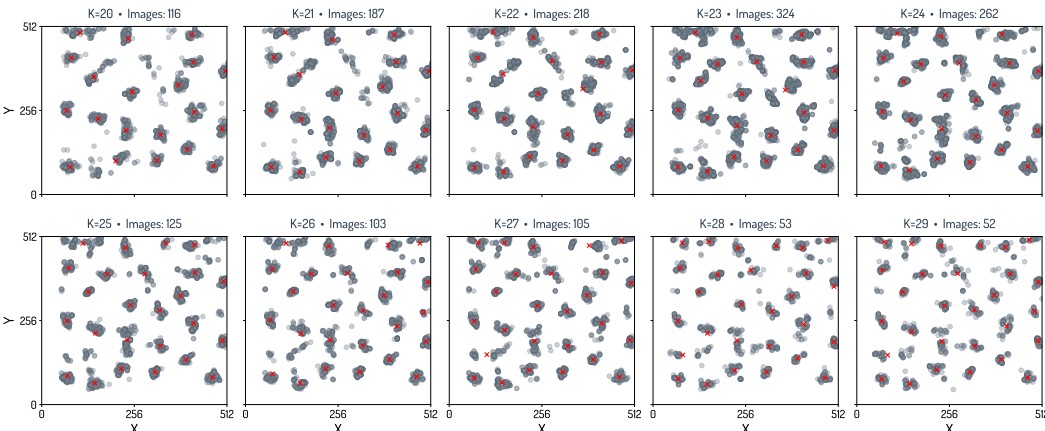

Figure 16: K-means clustering of object centers for actual counts 20–29 under a fixed noise prior. Each subplot corresponds to a specific actual count $n$, with dots representing object centers and red crosses denoting cluster centers. We focus on this range because the number of images is sufficient for reliable statistics, and the clustering structure is most evident. For very small or large $n$, the sample size is limited and the clustering is less clear.

**Stability Analysis.** As shown in Table 9, even with a small threshold $\tau = 0.05$, most cluster centers demonstrate remarkable stability with scores above 90%. When increasing the threshold to $\tau = 0.10$, almost all centers achieve perfect stability scores, with only minor variations at $n = 12$, $n = 26$, and $n = 27$. This high stability across different thresholds indicates that the spatial layout determined by noise prior is highly consistent between consecutive counts.

Table 9: Stability scores across different thresholds ($\tau$) and $n$ values. Values less than 1.000 are highlighted in *italic*.

| $\tau$ | $n = 11 \sim 20$ | | | | | | | | | | $n = 21 \sim 30$ | | | | | | | | | |
|---|---|---|---|---|---|---|---|---|---|---|---|---|---|---|---|---|---|---|---|---|
| | 11 | 12 | 13 | 14 | 15 | 16 | 17 | 18 | 19 | 20 | 21 | 22 | 23 | 24 | 25 | 26 | 27 | 28 | 29 | 30 |
| 0.05 | *.900* | *.909* | 1 | *.769* | *.929* | 1 | *.938* | 1 | *.944* | *.947* | 1 | 1 | *.955* | 1 | 1 | *.920* | *.846* | *.963* | 1 | 1 |
| 0.10 | 1 | *.909* | 1 | 1 | 1 | 1 | 1 | 1 | 1 | 1 | 1 | 1 | 1 | 1 | 1 | *.920* | *.885* | 1 | 1 | 1 |
| 0.15 | 1 | 1 | 1 | 1 | 1 | 1 | 1 | 1 | 1 | 1 | 1 | 1 | 1 | 1 | 1 | 1 | *.885* | 1 | 1 | 1 |
| 0.20 | 1 | 1 | 1 | 1 | 1 | 1 | 1 | 1 | 1 | 1 | 1 | 1 | 1 | 1 | 1 | 1 | *.923* | 1 | 1 | 1 |

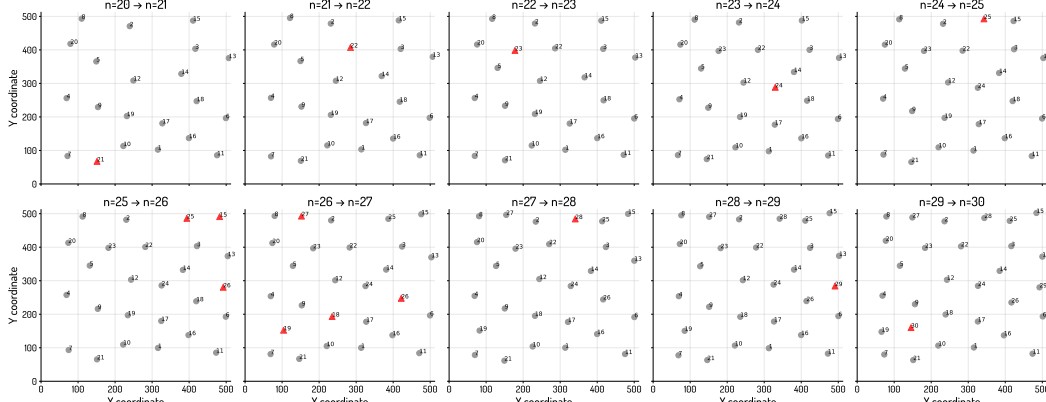

Figure 17: Sequential visualization of cluster center evolution from $n = 20$ to $n = 30$. Each panel shows the transition between consecutive counts, with gray circles indicating stable centers (displacement $< \tau$) and red triangles marking newly emerged centers. This step-by-step visualization demonstrates both the stability of existing centers and the structured emergence of new ones.

**Evolution Analysis.** To better understand how cluster centers evolve with increasing object count, we visualize the sequential changes from $n = 20$ to $n = 30$ using $\tau = 0.10$ as the matching threshold. Figure 17 presents this evolution as a series of snapshots, where each panel shows the transition from $n$ to $n + 1$ objects. Gray circles represent stable centers (with displacement less than $\tau$), while red triangles highlight newly emerged centers at each step. This sequential visualization reveals several interesting patterns: 1) The majority of existing centers remain highly stable across consecutive counts; 2) This systematic growth pattern suggests that the model has learned an implicit spacing rule for object placement while preserving the overall spatial arrangement.

# H TRAINING DYNAMICS AT DIFFERENT STEPS

This section provides a detailed view of training dynamics across different model architectures and dataset scales, complementing the scaling analysis in Section 4. While our main results in Figure 6 compare models at their optimal training steps (SD3.5-Medium at 3k steps, SD3.5-Large at 6k steps, and FLUX at 10k steps), here we present the complete training trajectories to illustrate how performance evolves throughout the training process.

The most striking observation across these training dynamics is the limited benefit from scaling training compute. Despite extending training to 10,000 steps and increasing dataset size to 100k examples, all models struggle to exceed 20% exact accuracy overall, with performance on counts above 30 showing minimal to no improvement:

- **FLUX** (Figure 19) shows a generally upward trend in accuracy metrics throughout training. Its Exact Accuracy increases more gradually than tolerance-based metrics. However, even after 10,000 steps, accuracy plateaus around 20%, with negligible gains on higher counts (>30).

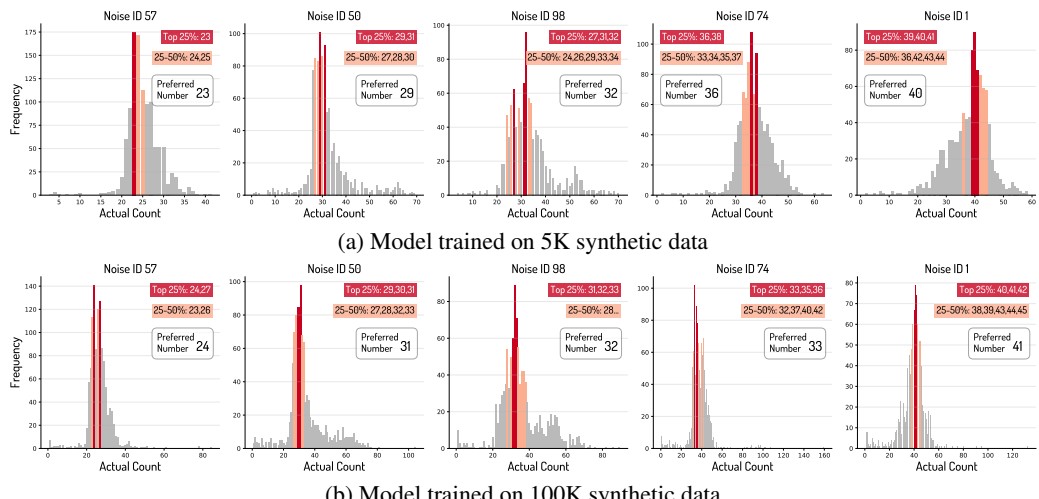

(a) Model trained on 5K synthetic data

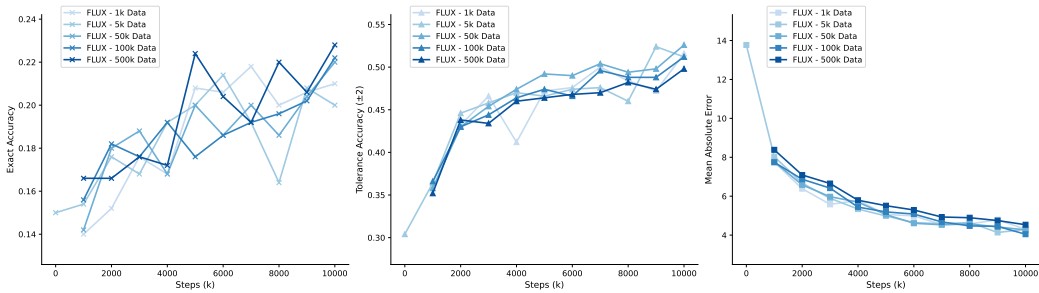

(b) Model trained on 100K synthetic data

Figure 18: Count distributions for the same set of noise priors using models trained on (a) 5K and (b) 100K synthetic data. For each noise prior, the preferred number and the overall distribution pattern are highly consistent across both models, despite the large difference in training data scale.

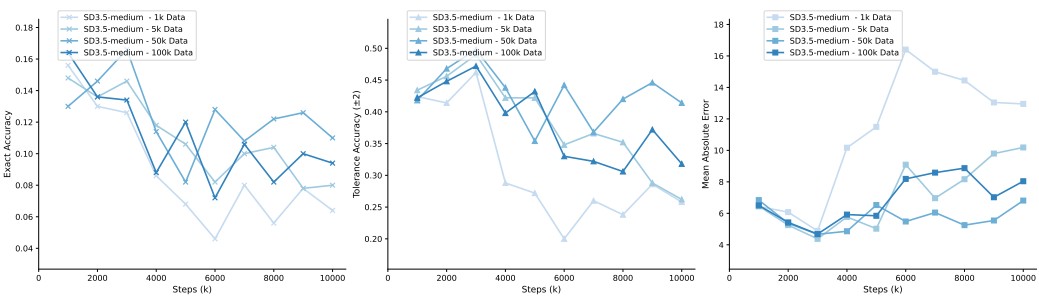

Figure 19: Training dynamics at different scales. Performance of FLUX when trained on 1k, 5k, 50k, and 100k examples. From left to right we plot Exact Accuracy (%↑), Tolerance Accuracy ($T=2$, %↑), and Absolute Error (↓) as a function of training steps.

- **SD3.5-Medium** (Figure 20) demonstrates earlier convergence but a lower performance ceiling, with diminishing returns beyond 3,000 steps.

These results strongly suggest that the limitations in numerosity generation are fundamentally architectural rather than a matter of insufficient data or training.

Figure 20: Training dynamics for SD3.5-Medium. Performance metrics when trained on different dataset scales, showing convergence around 3k steps.

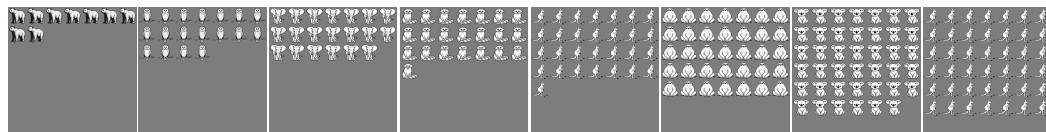

Figure 21: Illustration of the synthetic dataset based on grid layouts. We show grid-based images consisting of 9 black bears, 18 owls, 20 elephants, 22 raccoons, 29 kangaroos, 35 sloths, 41 koalas, and 42 kangaroos.

## I    ABLATION STUDIES

In this section, we conduct ablation study to analyze key factors influencing numerosity perception in diffusion models. We explore how spatial arrangements and noise prior settings impact counting accuracy to provide insights for practical implementations. All experiments use FLUX.1-dev model finetuned on our 5K synthetic dataset with consistent settings across evaluations.

**Layout Comparison.** We first investigate how spatial arrangement affects counting performance. Apart from random layout, we design another layout strategies: grid layout. As shown in Figure 21, objects are placed in a fixed $7 \times 7$ grid from top-left to bottom-right. The grid layout scheme ensures an easier counting task, as the optimization objective is reduced to rendering objects at a set of predefined positions for a given number. As shown in Table 11, all models achieve substantially better results under grid layout across all metrics. The structured grid arrangement simplifies counting by reducing spatial ambiguity, whereas random layout demands more complex reasoning about object locations during the diffusion process. Notably, even with such simplified spatial arrangements, exact accuracy for complex models like FLUX.1-dev remains below 40%, highlighting the persistent challenge of precise counting for diffusion models.

**Noise Prior Parameter Analysis.** We further examine how parameter choices in noise prior methods affect counting performance. As shown in Figure 23, all methods demonstrate clear sensitivity to parameter selection. Gaussian kernels work best with moderate weight ($\omega \approx 0.5$) and area proportion ($\alpha \approx 0.7$), while Uniform Scaled achieves optimal results with medium intensity adjustments ($\gamma \approx 0.5$). As visualized in Figure 22, there exists a clear trade-off: stronger noise modification yields higher counting accuracy but compromises image aesthetics. The Gaussian approach preserves better visual quality with modest accuracy gains, while Fixed and Uniform Scaled achieve precise counts but generate less natural objects.

**Image Quality Analysis** To further assess potential trade-offs between numerosity accuracy and image quality, we conducted a comparative evaluation of our Gaussian prior method against the baseline (training without noise prior). We generated 500 image pairs using identical prompts spanning counts 1–50, with each pair containing one image from each method. Quality assessment was performed through both human evaluation and automated analysis using Qwen2.5-VL-7B-Instruct. As shown in Table 10, the Gaussian prior method introduces no noticeable quality degradation.

Table 10: Image quality comparison between Gaussian prior and baseline

| Evaluation Method | Gaussian Better | Baseline Better | Tie |
|---|---|---|---|
| MLLM (Qwen2.5-VL) | 8.0% | 4.6% | 87.4% |
| Human Evaluation | 30.0% | 23.3% | 46.7% |

These ablation studies further corroborate our core finding that noise priors fundamentally constrain numerosity perception, while also providing practical guidelines for optimizing diffusion models' counting abilities.

## J    EXTENDED ANALYSIS OF GENERALIZATION CAPABILITIES

This section provides comprehensive experiments addressing the generalization of our noise-prior methods beyond the main experimental settings, including extended count ranges and varied object size conditions.

Figure 22: Comparison of images generated with different local noise priors. The figure shows results for 7 objects (left column) and 32 objects (right column). Moving from left to right, count accuracy progressively improves; while moving from right to left, aesthetic quality gradually increases.

Table 11: Model performance comparison across spatial arrangements. Results show all three models achieve significantly better numerosity perception with grid layout across all metrics. Best performance in each column is highlighted in **bold**.

| Model | Exact Accuracy (%) | | Mean Absolute Error | | Tolerance Accuracy (%) | |
|---|---|---|---|---|---|---|
| | random | grid | random | grid | random | grid |
| SD-3.5-M | 18.4 | 41.2 | 4.36 | 1.05 | 50.4 | 89.0 |
| SD-3.5-L | 18.8 | **52.7** | **2.73** | **0.72** | **54.0** | **93.7** |
| FLUX.1-dev | **20.0** | 35.5 | 4.30 | 1.20 | 51.2 | 84.9 |

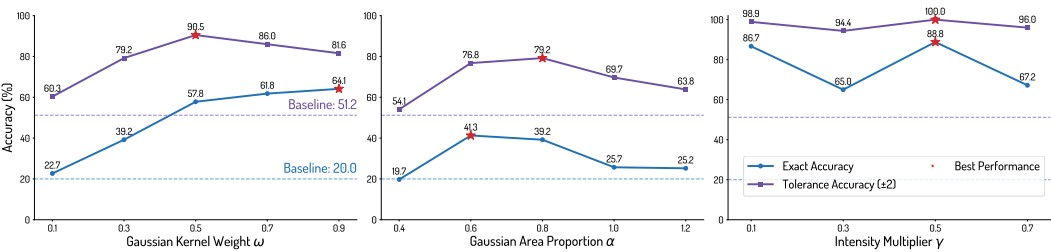

Figure 23: Parameter sensitivity analysis for noise prior methods. Left: Gaussian kernel weight ($\omega$) performance peaks at 0.5. Center: Area proportion factor ($\alpha$) works best within 0.6-0.8. Right: Intensity multiplier ($\gamma$) optimal at 0.5. Dashed lines show baseline performance.

## J.1 GENERALIZATION TO HIGHER OBJECT COUNTS (1–100)

To evaluate whether our conclusions generalize beyond the 1–50 range studied in the main paper, we extended training and evaluation to 1–100 objects. We generated 10K samples following our original pipeline and fine-tuned FLUX on this extended range, evaluating on held-out object categories, where baseline means training without noise priors.

The results demonstrate two key findings: (1) Baseline performance degrades significantly beyond 60 objects, with near-zero exact accuracy at higher counts, confirming that diffusion models' difficulty with numerosity scales with object count; (2) Our uniform scaled prior maintains robust performance across all ranges, achieving 53.8% overall exact accuracy and maintaining over 40% tolerance accuracy even in the challenging 90–100 range. This shows that our noise-based approach generalizes effectively to higher numerosities.

## J.2 ROBUSTNESS TO OBJECT SIZE VARIATION

We first analyze the correlation between object area and count using GroundingDINO to extract layout information. The moderate correlations (training: 0.447, validation: 0.413) suggest that area is not a strong predictor of count in our dataset, providing initial evidence against size-based shortcut learning. To further investigate whether models rely on object size or total area as shortcuts for numerosity estimation, we conducted experiments under varying object size conditions.

Table 12: Performance comparison on extended count range (1–100 objects)

| Method | Exact Accuracy by Count Range (%) | | | | | | | | | | Overall |
|---|---|---|---|---|---|---|---|---|---|---|---|
| | 1-10 | 10-20 | 20-30 | 30-40 | 40-50 | 50-60 | 60-70 | 70-80 | 80-90 | 90-100 | |
| Baseline | 63.8 | 17.0 | 7.0 | 6.0 | 6.0 | 2.0 | 0.0 | 1.0 | 1.0 | 0.0 | 9.2 |
| Uniform Scaled | **88.9** | **97.0** | **97.0** | **95.0** | **64.0** | **38.4** | **28.0** | **18.0** | **7.0** | **11.5** | **53.8** |

Table 13: Tolerance accuracy (T=2) on extended count range

| Method | Tolerance Accuracy by Count Range (%) | | | | | | | | | | Overall |
|---|---|---|---|---|---|---|---|---|---|---|---|
| | 1-10 | 10-20 | 20-30 | 30-40 | 40-50 | 50-60 | 60-70 | 70-80 | 80-90 | 90-100 | |
| Baseline | 98.8 | 78.0 | 49.0 | 24.0 | 26.0 | 19.0 | 7.0 | 4.0 | 1.0 | 2.0 | 29.2 |
| Uniform Scaled | **98.9** | **100.0** | **100.0** | **100.0** | **100.0** | **93.9** | **77.0** | **65.0** | **49.0** | **42.7** | **82.2** |

**Baseline Performance with Size Variation** As shown in Table 14, the comparable performance (22.8% vs. 20.0% overall accuracy) further confirms that the model does not primarily rely on object size or total area as shortcuts for numerosity estimation.

Table 14: Baseline performance with identical vs. varying object sizes

| Training Setting | Exact Accuracy by Count Range (%) | | | | Overall |
|---|---|---|---|---|---|
| | 1-10 | 10-20 | 20-30 | >30 | |
| Identical Size | 77.8 | 19.0 | 3.00 | 3.81 | 20.0 |
| Varying Size | 75.6 | 25.0 | 9.00 | 5.71 | 22.8 |

**Noise Prior Method with Size Variation** As shown in Table 16, our noise prior method maintains strong performance under varying size conditions (83.8% vs. 85.3% overall), demonstrating consistent layout control ability regardless of object size uniformity. More importantly, when compared to the baseline performance with size variation in Table 14 (22.8% overall), our method achieves significantly higher accuracy (83.8% overall), confirming that noise prior manipulation remains highly effective even when object sizes vary.

## J.3 IMPACT OF TRAINING CATEGORY DIVERSITY ON GENERALIZATION

To assess whether our findings rely on category-specific memorization, we conducted controlled experiments using varying numbers of object categories during training. As shown in Table 15, when training on a single category ('elephant'), performance on held-out categories ('rabbit') remains comparable to within-category evaluation (22.0% vs. 24.2% exact accuracy). Moreover, expanding to 20 categories yields similar held-out accuracy (21.0%), indicating that numerosity learning is largely category-agnostic. These results confirm that the observed limitations stem from fundamental difficulties with spatial-numerical mapping rather than domain-specific overfitting.

These experiments collectively demonstrate that our methods and findings generalize to higher object counts and varied size conditions, confirming that the observed phenomena are not artifacts of specific experimental settings but reflect fundamental properties of diffusion models for numerosity tasks.

## K UNSUCCESSFUL ATTEMPTS TO REDUCE NOISE BIAS

Beyond explicit noise prior injection, we explored several alternative strategies to mitigate the inherent bias in noise initialization. While these approaches showed some promise, the improvements were modest. We organize our exploration into two categories: training-time modifications and inference-time adjustments.

Table 15: Impact of training category diversity on numerosity generalization

| Training Objects | Eval Objects | Exact Acc (%) | Tolerance Acc (%) | MAE |
|---|---|---|---|---|
| Single ('elephant') | 'elephant' (seen) | 24.2 | 57.4 | 3.68 |
| Single ('elephant') | 'rabbit' (held-out) | 22.0 | 51.0 | 4.64 |
| 20 categories | 'rabbit' (held-out) | 21.0 | 51.6 | 4.30 |

Table 16: Noise prior performance with identical vs. varying object sizes

| Training Setting | Exact Accuracy by Count Range (%) | | | | Overall |
|---|---|---|---|---|---|
| | 1-10 | 10-20 | 20-30 | >30 | |
| Identical Size | 87.8 | 94.9 | 96.2 | 75.0 | 85.3 |
| Varying Size | **98.9** | **97.0** | 93.0 | 66.7 | 83.8 |

### K.1 TRAINING-TIME MODIFICATIONS

We investigated two approaches to reduce noise dependency during training: Early Timestep Sampling and Shared Noise Batching.

**Early Timestep Sampling** This approach modifies the timestep sampling distribution from uniform to a logit-normal distribution, allocating more compute to earlier denoising steps where layout formation occurs. Table 17 shows performance across different logit mean values (with fixed logit std=1.0). Negative logit means, particularly $\mu = -1.5$, demonstrate moderate improvements (+5.0% in exact accuracy), suggesting that emphasizing early denoising steps helps the model establish better object layouts. However, despite this targeted compute reallocation, the overall benefits remain limited.

**Shared Noise Batching** This technique uses one noise vector shared across all samples in a batch, forcing the model to rely more on text instructions than noise patterns. It provides only a slight improvement (+0.4% in exact accuracy) over the baseline, suggesting that while shared noise may reduce some noise-specific biases, it does not fundamentally alter the model's dependence on noise for spatial arrangement.

### K.2 INFERENCE PARAMETER ADJUSTMENTS

We examined two parameter modifications during inference to assess their impact on counting performance: classifier-free guidance scale tuning and sampling batch size adjustment inspired by previous work focus on inference-time scaling Muennighoff et al. (2025); Snell et al. (2024); Ma et al. (2025).

**Classifier-Free Guidance** We modified the CFG scale to balance adherence to the text prompt versus the noise prior. As shown in Table 18, adjusting CFG values yields only marginal improvements. The small variance in performance (+2.2% in tolerance accuracy at best) indicates that CFG tuning alone cannot overcome the fundamental noise-driven layout biases.

**Sampling Batch Size** We varied the number of sampling iterations through increased batch sizes to determine if allowing the model to explore a wider range of solutions could overcome noise bias. Surprisingly, as shown in Table 19, increased sampling iterations with our Gaussian noise prior method actually reduced performance slightly. This suggests that the underlying noise bias is consistent across sampling attempts, and additional sampling primarily introduces variance rather than improving accuracy.

## L LIMITATIONS & FUTURE WORK

We raise several important questions for future work. *Why do diffusion models tend to rely on 2D spatial noise priors rather than textual prompt instructions?* We anticipate that one key reason may lie in the theoretical limitations of diffusion models—specifically, that their optimization target is the given 2D noise prior. *Can autoregressive image generation models ensure accurate numerosity?* The

Table 17: Performance comparison with different Early Timestep Bias (logit mean values).

| Logit Mean | Exact Accuracy (%) | Tolerance Accuracy (%) | Mean Absolute Error |
|---|---|---|---|
| -1.5 | 25.0 | 54.4 | 3.46 |
| -1.0 | 20.8 | 52.0 | 4.01 |
| 0.0 (Uniform) | 20.0 | 51.2 | 4.30 |
| 1.0 | 16.2 | 36.8 | 11.6 |

Table 18: Performance with different classifier-free guidance scale values.

| CFG Scale | Exact Accuracy (%) | Tolerance Accuracy (%) | Mean Absolute Error |
|---|---|---|---|
| 2.0 | 19.4 | 48.8 | 4.99 |
| 3.5 (Default) | 20.0 | 51.2 | 4.30 |
| 5.0 | 19.8 | 52.2 | 4.26 |
| 7.0 | 18.4 | 50.2 | 4.82 |

answer is no at the current stage, as we have demonstrated that even the most advanced model, GPT-4o, still struggles with accurate numerosity. *Are there any fundamental tasks that scaling diffusion models still fail to address?* We anticipate that spatial control may be equally challenging.

**LLM Usage Statement** We used large language models solely to aid in polishing the writing of this paper, including improving grammar, wording, and sentence clarity. All technical content, research ideas, experimental results, and scientific claims are entirely our own. The models were not used for any research ideation, data analysis, or code generation. We have carefully verified all content to ensure accuracy and adherence to academic standards.

Table 19: Effect of increased sampling iterations using Gaussian noise prior.

| Sampling Strategy | Exact Accuracy (%) | Tolerance Accuracy (%) | Mean Absolute Error |
|---|---|---|---|
| Default (500 samples) | 39.2 | 79.2 | 1.54 |
| BS=10 (2,500 samples) | 36.6 | 75.1 | 1.68 |
| BS=50 (12,500 samples) | 37.0 | 76.1 | 1.63 |

