# OpenReview forum: "Demystifying Numerosity in Diffusion Models — Limitations and Remedies"
_ICLR.cc/2026/Conference — ICLR 2026 Conference Withdrawn Submission_

### Official Review · Reviewer_YYgC · 2025-10-24

**Soundness:** 3
**Presentation:** 2
**Contribution:** 2
**Rating:** 4
**Confidence:** 3

**Summary:**

This work investigates the challenge of diffusion models to generate the exact number of objects specified in a prompt. The authors' findings indicate that this issue cannot resolved by fine-tuning or employing larger model sizes. Furthermore, they demonstrate that the generated object count is highly dependent on the sampled noise prior rather than the specified prompt. This suggests the noise prior often "overwrites" the desired count. Finally, the authors propose several simple methods to improve the quality of the sampled noise.

**Strengths:**

- This paper contributes to understanding why diffusion models produce the incorrect count through intuitive experiments, addressing a significant issue in the field.

- The authors provide synthetic datasets specifically for training and testing counting performance.

- They propose simple noise conditioning techniques that can help improve counting accuracy.

**Weaknesses:**

- Existing work, such as ReNO, has already explored the effect of tuning the initial noise of diffusion models and includes results for object counting. However, this paper neither cites nor explains or compares itself to such related work.

- The Related Work section mentions three paths for achieving numeracy but does not elaborate on them.

- Generally, it remains somewhat unclear how the prior manipulations (uniform, fixed, Gaussian) were chosen and precisely why they are effective.

**Questions:**

- For the fixed prior type, how is $z^*$ determined?

---

### Official Review · Reviewer_1uYR · 2025-10-29

**Soundness:** 3
**Presentation:** 3
**Contribution:** 3
**Rating:** 6
**Confidence:** 3

**Summary:**

This paper shows that state-of-the-art text-to-image diffusion models systematically miscount objects requested by a prompt. The authors build two synthetic benchmarks, GrayCount250 (gray backgrounds with 1 to 50 objects) and NaturalCount6 (naturalistic scenes with 2 to 6 objects), to enable rigorous evaluation. Scaling model capacity from 2B to 12B parameters and training data from 1K to 500K samples does not improve counting accuracy. Analysis indicates the generators rely primarily on the initial noise rather than text conditioning to set spatial layout and numerosity, and different noise priors favor different object counts. To address this, the authors introduce count-aware noise conditioning that injects layout information into the noise prior, raising accuracy from 20.0% to 85.3% on GrayCount250 and from 74.8% to 86.3% on NaturalCount6.

**Strengths:**

1. Rigorous experimental design with high-quality synthetic datasets. The construction of GrayCount250 and NaturalCount6 addresses a fundamental challenge in numerosity research—obtaining reliable ground-truth annotations. The transparent object generation pipeline with paste-layer-to-layout composition enables scalable data synthesis while ensuring count accuracy. The evaluation methodology using CountGD for automated counting (achieving 99.78% exact accuracy on held-out categories, Table 7) provides a solid foundation for reproducible benchmarking.
2. Novel insights into noise prior dominance with compelling empirical evidence. The discovery that different noise initializations determine distinct counting ranges (Figures 8, 13-15) represents a significant finding about diffusion model behavior. The statistical analysis of spatial patterns through K-means clustering (Figure 16) and stability metrics (Table 9) provides strong quantitative support. The cross-model consistency analysis (Figure 18) demonstrates this phenomenon persists across different training scales, suggesting a fundamental architectural limitation rather than a data-dependent artifact.
3. Nice and clear visualization. The paper excels in visual presentation across multiple dimensions. Figure 9's side-by-side comparison clearly demonstrates how fixed noise produces consistent layouts across different prompts and counts, directly supporting the "noise determines layout, text activates locations" hypothesis. The count distribution histograms (Figures 8, 13-15) with overlaid statistics make the preferred numerosity modes immediately apparent.

**Weaknesses:**

1. Limited evaluation on naturalistic images undermines generalization claims. While GrayCount250 covers counts 1-50, the naturalistic NaturalCount6 dataset only spans 2-6 objects, as stated in Section 3.2: "We construct a naturalistic dataset focusing on counts from 2 to 6 objects, where real-world annotations are particularly challenging due to occlusion and clutter." This narrow range is insufficient to validate whether findings about noise prior dominance and the failure of scaling extend to higher counts in complex real-world scenarios.

2. Limited exploration of non-noise interventions before concluding noise manipulation is necessary. Appendix K labels alternatives as “modest,” but the evaluation is incomplete. Table 17 shows early-timestep sampling (logit mean −1.5) reaches 25% accuracy, a five-point gain over the 20% baseline. The paper does not test synergy with noise-prior methods, probe other timestep distributions or schedulers, or assess learnable schedules. At minimum, ablate combinations with noise priors, sweep alternative sampling distributions, and evaluate learnable or data-conditioned schedules.

3. (Minor) Missing discussion of several related works on the counting abilities of visual generative models. While the literature review is comprehensive and the proposed method is effective, there are additional studies on counting abilities that may require discussion, particularly those examining different visual model architectures [1,2,3] and varied settings (e.g., world knowledge [4] or broad, nonspecific benchmarks [5]). This is a suggestion for improvement and not a basis for rejection.

**Questions:**

1. Can you provide evidence (even preliminary results on a small dataset) showing that the same noise seeds exhibit consistent counting preferences in naturalistic scenes with counts >6?

2. Is it due to fundamental differences in training data distribution, evaluation methodology differences between GeckoNum and your benchmark, or something lost during fine-tuning that the pretrained model possessed?

3. Have you analyzed attention maps to quantify how much spatial information flows from noise features versus text embeddings at different timesteps?

### References

[1] Roni Paiss, Ariel Ephrat, Omer Tov, Shiran Zada, Inbar Mosseri, Michal Irani, Tali Dekel. “Teaching CLIP to count to ten”. ICCV 2023.

[2] Xuyang Guo, Zekai Huang, Jiayan Huo, Yingyu Liang, Zhenmei Shi, Zhao Song, Jiahao Zhang. “Can you count to nine? A human evaluation benchmark for counting limits in modern text-to-video models”. arXiv 2025.

[3] Xuyang Guo, Zekai Huang, Zhenmei Shi, Zhao Song, Jiahao Zhang. “Your Vision-Language Model Can't Even Count to 20: Exposing the Failures of VLMs in Compositional Counting”. arXiv 2025.

[4] Yubin Chen, Xuyang Guo, Zhenmei Shi, Zhao Song, Jiahao Zhang. “T2VWorldBench: A Benchmark for Evaluating World Knowledge in Text-to-Video Generation”. WACV 2026.

[5] Vitali Petsiuk, Alexander E. Siemenn, Saisamrit Surbehera, Zad Chin, Keith Tyser, Gregory Hunter, Arvind Raghavan, Yann Hicke, Bryan A. Plummer, Ori Kerret, Tonio Buonassisi, Kate Saenko, Armando Solar-Lezama, Iddo Drori. “Human evaluation of text-to-image models on a multi-task benchmark”. arXiv 2022.

---

### Official Review · Reviewer_vZpi · 2025-11-02

**Soundness:** 3
**Presentation:** 3
**Contribution:** 2
**Rating:** 4
**Confidence:** 4

**Summary:**

This paper examines whether modern text-to-image diffusion models can reliably follow numerosity instructions. The authors introduce two reproducible evaluation sets—GrayCount250 (synthetic) and NaturalCount6 (natural scenes)—to measure counting accuracy. They further propose injecting count-aware layout information into the initial noise prior and report substantial gains on both datasets.

**Strengths:**

1. Numerosity is a persistent failure mode for T2I models; a focused study is valuable to the community. The proposed method shows large improvements on counting metrics.
2. Propose two datasets covering synthetic and real-world scenarios, enabling systematic and quantitative analysis.

**Weaknesses:**

1. **Limited novelty / positioning:** The core technique modifies the initial noise to guide multi-object generation. Ban et al. [1] similarly inject or modify noise patches (Sec. 3.2) and observe improved multi-object placement (App. G). Beyond targeting larger counts, it’s not yet clear how this work is distinct in mechanism or capability. Please clarify conceptual and technical differences.
2. **Generalization:**
   a) **Model coverage:** Section 5 evaluates only FLIX.1-dev. Please include additional models with different backbones/conditioning to test robustness.
   b) **Sampler diversity:** What sampler is used in Section 5? If deterministic, please also evaluate a **stochastic** sampler (e.g., with noise schedules or ancestral steps) to test stability under sampling variability.  I believe it will distort the initial noise a lot.
3. **Baselines:** Please compare against (i) noise-based methods such as Ban et al. [1] and Guo et al.[3] and (ii) attention-editing methods such as **Attend-and-Excite** [2] under the same prompts and metrics.

[1] Ban, Y., Wang, R., Zhou, T., Gong, B., Hsieh, C. J., & Cheng, M. (2024). *The Crystal Ball Hypothesis in diffusion models: Anticipating object positions from initial noise.* arXiv:2406.01970.
[2] Chefer, H., Alaluf, Y., Vinker, Y., Wolf, L., & Cohen-Or, D. (2023). *Attend-and-Excite: Attention-based semantic guidance for text-to-image diffusion models.* TOG 42(4).
[3] Guo, X., Liu, J., Cui, M., Li, J., Yang, H., & Huang, D. (2024). Initno: Boosting text-to-image diffusion models via initial noise optimization. In Proceedings of the IEEE/CVF Conference on Computer Vision and Pattern Recognition (pp. 9380-9389).

**Questions:**

1) What’s the design intuition behind the Table 2 noise patterns, and how sensitive are results to these choices?
2) Can you quantify aesthetic impacts (e.g., HPSv2, CLIP-IQA, SSIM/LPIPS, or user studies), given [1]’s reported degradations?

---

### Official Review · Reviewer_KVQs · 2025-11-02

**Soundness:** 2
**Presentation:** 2
**Contribution:** 2
**Rating:** 2
**Confidence:** 4

**Summary:**

This paper examines why text-to-image diffusion models struggle with numerosity—following numerical instructions such as “five apples.” The authors introduce two benchmarks, GrayCount250 (synthetic geometric shapes, 1–50 objects) and NaturalCount6 (simple natural images with 2–6 objects). They empirically observe that scaling model or dataset size does not improve counting, and that the initial noise seed largely determines object count and spatial arrangement. To mitigate this, they propose “count-aware noise priors,” injecting structured noise patterns that correspond to the desired number of objects, yielding higher count accuracy on both benchmarks.

**Strengths:**

Clean, reproducible experiments. The datasets are straightforward, and the analysis pipeline is transparent and well documented.

Simple implementation. The proposed noise prior is easy to apply and clearly demonstrates its effect.

Readable and polished. The paper is well written, and the figures communicate results effectively.

**Weaknesses:**

The main finding—that noise initialization strongly drives spatial layout and thus object count—is not new. Prior analyses (e.g., Li et al., 2025) already demonstrate that early noise determines coarse composition and that text guidance only refines details. This work re-affirms that behavior for counting but does not deepen our understanding or propose a new mechanism.

The proposed “count-aware noise prior” is another instance of layout conditioning: encoding the desired count geometrically through structured noise. Such layout-based interventions have been studied in multiple diffusion-control settings. The method improves accuracy because it injects the correct layout, not because it teaches the model numerical reasoning.
[1] Detection‑Driven Object Count Optimization for Text‑to‑image Diffusion Models
[2] Make It Count
[3] CountDiffusion
[4] CountCluster
[5] Training‑Free Layout Control with Cross‑Attention Guidance
[6] LayoutDiffusion

Both benchmarks are highly simplified—geometric shapes or small objects in clean backgrounds with literal numeric prompts. There is no evidence the method generalizes to open-domain, natural-language prompts or real scenes with occlusion and scale variation.

The claim that noise “dominates text conditioning” is supported only by qualitative correlation (same noise → same count). The paper does not analyze attention strength or conditioning gradients to demonstrate causal dominance. The conclusions remain observational.

Terms like “demystifying” and “disproving the scaling hypothesis” overstate the novelty. It is now empirically well established—across diffusion and large-language models—that scaling alone does not endow symbolic reasoning or compositional accuracy. The presented results are incremental, not paradigm-shifting.

**Questions:**

Several existing works (e.g., LayoutDiffusion, NoiseCollage, LoCo, Make-It-Count) already use structured or layout-aware noise to control object placement and count. What is fundamentally new about “count-aware noise prior” beyond being another form of structured initialization?

The paper claims “noise dominance” over text conditioning but provides only correlational evidence. Can the authors show any causal or quantitative analysis (e.g., attention strength, gradient tracing) proving that noise truly overrides text signals rather than simply co-determining layout?

Results are limited to simple synthetic datasets. Does the method still work for real-world prompts with varied object scales, occlusion, or approximate quantifiers (“a few,” “several”)?

---

### Note · Authors · 2025-11-30

I have read and agree with the venue's withdrawal policy on behalf of myself and my co-authors.